# Conditional Outcome Equivalence: A Quantile Alternative to CATE

**Josh Givens**
University of Bristol
josh.givens@bristol.ac.uk

**Henry W J Reeve**
University of Bristol
henry.reeve@bristol.ac.uk

**Song Liu**
University of Bristol
song.liu@bristol.ac.uk

**Katarzyna Reluga**
University of Bristol
katarzyna.reluga@bristol.ac.uk

## Abstract

The conditional quantile treatment effect (CQTE) can provide insight into the effect of a treatment beyond the conditional average treatment effect (CATE). This ability to provide information over multiple quantiles of the response makes the CQTE especially valuable in cases where the effect of a treatment is not well-modelled by a location shift, even conditionally on the covariates. Nevertheless, the estimation of the CQTE is challenging and often depends upon the smoothness of the individual quantiles as a function of the covariates rather than smoothness of the CQTE itself. This is in stark contrast to the CATE where it is possible to obtain high-quality estimates which have less dependency upon the smoothness of the nuisance parameters when the CATE itself is smooth. Moreover, relative smoothness of the CQTE lacks the interpretability of smoothness of the CATE making it less clear whether it is a reasonable assumption to make. We combine the desirable properties of the CATE and CQTE by considering a new estimand, the conditional quantile comparator (CQC). The CQC not only retains information about the whole treatment distribution, similar to the CQTE, but also having more natural examples of smoothness and is able to leverage simplicity in an auxiliary estimand. We provide finite sample bounds on the error of our estimator, demonstrating its ability to exploit simplicity. We validate our theory in numerical simulations which show that our method produces more accurate estimates than baselines. Finally, we apply our methodology to a study on the effect of employment incentives on earnings across different age groups. We see that our method is able to reveal heterogeneity of the effect across different quantiles.

## 1 Introduction

In many real world scenarios such as personalised treatment allocation and individual level policy decisions, understanding the effect of a treatment/intervention at an individual level is invaluable in providing bespoke care. The field which aims to understand a treatment's effect given certain covariates is referred to as heterogeneous treatment effect (HTE) estimation and has seen popularity across many applications [11, 10, 24, 20]. Within HTE, the conditional average treatment effect (CATE) has proved itself to be a popular target of study in this area due to its simplicity and interpretability [2, 13, 28]. A key limitation of the CATE however, is that it fails to paint a full picture of the differences between distributions of the two responses. In addition, it can be sensitive to outliers, with extreme values leading to a biased outcome. As such, the conditional quantile treatment effect (CQTE), an estimand which compares the conditional quantiles of the distributions in the treated and untreated populations, has established itself as a popular alternative [1, 5, 26].

38th Conference on Neural Information Processing Systems (NeurIPS 2024).

While the CQTE offers more information than the CATE and is more robust to outliers, it lacks some of the CATE's desirable estimation properties. Specifically, CQTE estimation involves estimating the quantile functions for the two marginal outcomes. This harms the estimation procedure in cases where estimation of marginal quantile functions is more challenging than estimation of the CQTE itself. An example of this is when the marginal quantile functions are less smooth as a function of the covariates than the CQTE. This aligns with a recurring idea within HTE estimation that the effect of a treatment may be simpler than the marginal outcomes. In contrast to the CQTE, there are many CATE estimators which aim to learn the CATE directly allowing them to exploit its relative simplicity. These include the X-learner [17], R-learner [23], and Doubly Robust (DR) learner [15]. Before estimating the CATE, these procedures require estimation of intermediary estimands (nuisance parameters) which condition on the covariates such as the average marginal outcomes and the propensity score (the probability of being assigned to treatment group). These nuisance parameters are then used to aid the estimation of the CATE. With the DR learner specifically, it has been shown that it can still achieve optimal convergence rates even when estimation of the nuisance parameters is worse than estimation of the CATE itself. This notion is referred to as **double robustness**, as our estimation is robust to sub-optimal estimation of both of the nuisance parameters.

Some attempts have been made to improve CQTE estimation [34, 33] with a key work being that of Kallus and Oprescu [14]. In this they provide an extension of the double robustness property to the CQTE, creating an estimation procedure that can achieve strong convergence even when nuisance parameters are more difficult to estimate. Unfortunately, one of the nuisance parameters which must be estimated is the reciprocal of the conditional densities over the response. These are highly difficult to estimate and risk the errors blowing up in low density regions which could potentially nullify the desirable estimation rates they achieve even with the dependence on the estimation accuracy of these nuisance parameters being less strong. Furthermore it is still unclear how one can interpret relative smoothness in the CQTE compared to the individual quantiles with their being relatively little discussion of this within the literature. In general there is a distinct lack of illustrative examples; which are present for the CATE. To our knowledge no other works specifically aim to tackle this double robustness phenomenon for the CQTE or other quantile based treatment effect estimands.

We introduce a novel estimand called the "conditional quantile comparator" (CQC). The CQC gives the outcome for a treated individual in the same quantile as a given outcome for an untreated individual, conditional on covariates. This relates to the conditional Quantile-Quantile (QQ) plot for the treated and untreated outcomes as demonstrated in Figure 1. Similarly to the CQTE, our new esti-

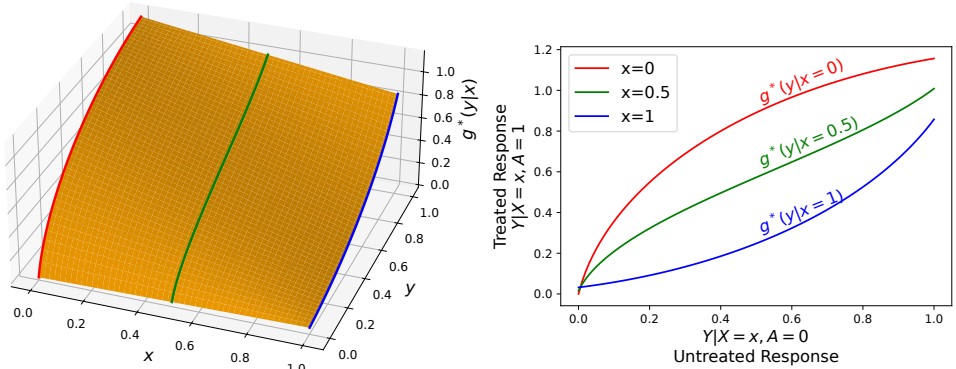

Figure 1: The left plot gives the CQC surface which takes in covariates $(x)$ and an untreated response $(y)$ and returns the treated response of the equivalent quantile $(g^*(y|x))$. The right plot is a QQ-plot of the responses $(Y)$ in the untreated $(A = 0)$ vs treated $(A = 1)$ population conditional on various covariates $(X = 0, 0.5, 1)$. These conditional QQ-plots correspond to "slices" of the CQC surface, as shown by the coloured lines in the left plot. The plot is best viewed in colour.

mand, the CQC, allows us to compare equivalent quantiles while working exclusively in the response landscape, making it a more interpretable tool. This allows us to construct canonical examples of the CQC being smoother than various nuisance parameters, the CQTE, and the CATE; adding to this interpretability. In addition, using the pseudo-outcome framework presented in Kennedy [15], we can leverage CATE estimation procedures to estimate the CQC in a doubly robust way, as mentioned

above. Crucially, the CQC can keep the valuable quantile-level information previously offered by the CQTE while building on much of the CATE literature to acquire its desirable robustness properties and interpretability. Our contributions are as follows:

- Introduce a new estimand for HTE analysis: the conditional quantile comparator (CQC).

- Propose an estimation procedure which we prove to be doubly robust.

- Demonstrate better estimation accuracy especially when the CQC is smooth but individual conditional cumulative distribution functions are not.

- Provide insights into real-world datasets on employment intervention and medical treatment.

## 2  Set-up

We now introduce the standard HTE set-up in our notation. Let $Z$ denote the random triple $(Y, X, A)$ with $Y$ a random variable (RV) on $\mathcal{Y} \subseteq \mathbb{R}$, $X$ a RV on $\mathcal{X} \subseteq \mathbb{R}^d$, and $A$ a RV on $\{0, 1\}$. We treat $Z$ as representing an individual and interpret the components as

$$Y : \text{Outcome/Response} \qquad X : \text{Observed covariates} \qquad A : \text{Treatment assignment}$$

**Remark 1.** *We could view our setting as coming from a potential outcome framework [27]. Under this framework we assume there exists RVs $Y_0, Y_1$ on $\mathcal{Y}$ representing the outcome with and without treatment and that $Y \equiv Y_A$. $Y_{1-A}$ would then be unobserved/unknown for each individual.*

We define the *propensity score* $\pi : \mathcal{X} \to (0, 1)$ by

$$\pi(\boldsymbol{x}) := \mathbb{P}(A = 1 | X = \boldsymbol{x}),$$

in other words, $\pi$ denotes the conditional probability of being assigned to treatment given the covariates. We shall assume that $\pi$ is continuous and bounded away from $0$ and $1$. From a potential outcomes perspective, this means that each individual could potentially be assigned to either treatment. We also define

$$F_a(y|\boldsymbol{x}) := \mathbb{P}(Y \le y | X = \boldsymbol{x}, A = a), \tag{1}$$

$$F_a^{-1}(\alpha|\boldsymbol{x}) := \inf\{y \in \mathbb{R} | F_a(y|\boldsymbol{x}) \ge \alpha\}, \tag{2}$$

and refer to them as the *conditional cumulative distribution function (CCDF)* and the *quantile function* respectively. We also refer to $F_a^{-1}$ in (2) as the *generalised inverse* of $F_a$.

We can now define the CATE and CQTE to be given by $\tau : \mathcal{X} \to \mathbb{R}$ and $\tau_q : [0, 1] \times \mathcal{X} \to \mathbb{R}$ with

$$\tau(\boldsymbol{x}) := \mathbb{E}[Y | X = \boldsymbol{x}, A = 1] - \mathbb{E}[Y | X = \boldsymbol{x}, A = 0],$$

$$\tau_q(\alpha|\boldsymbol{x}) := F_1^{-1}(\alpha|\boldsymbol{x}) - F_0^{-1}(\alpha|\boldsymbol{x}).$$

We let $D := \{Z_i\}_{i=1}^{2n} \equiv \{(Y_i, X_i, A_i)\}_{i=1}^{2n}$ for $n \in \mathbb{N}$ be IID copies of $Z$ representing our data sample with $i$ indexing the individual. We assume an even number of samples for notational convenience. For $a \in \{0, 1\}$, we take $I_a := \{i | A_i = 1\}$, the indices of individuals on treatment $a$. We can then define $D_a := \{Z_i\}_{\{i \in I_a\}}$ and $n_a := |I_a|$ as the dataset and sample size of those on treatment $a$.

For $n \in \mathbb{N}$, let $[n] := \{1, \ldots, n\}$. For a vector $\boldsymbol{w} \in \mathbb{R}^p$ let $w_j$ to represent the $j^{\text{th}}$ component of $\boldsymbol{w}$ and let $\|\boldsymbol{w}\|$ be the Euclidean norm unless otherwise specified. We also take $\|\boldsymbol{w}\|_1$ as the 1-norm and $\|\boldsymbol{w}\|_\infty := \max_{j \in [p]} |w_j|$. We keep a summary table of all notation used in Appendix A.1.

### 2.1  Introducing the quantile comparator

Our aim is to find "equivalent quantiles" between the treated and non-treated distributions conditional on the covariates. Specifically, for each $y_0 \in \mathcal{Y}$, $\boldsymbol{x} \in \mathcal{X}$ we aim to find $y_1$ such that

$$F_1(y_1|\boldsymbol{x}) = F_0(y_0|\boldsymbol{x}).$$

This now allows us to define our primary estimand of interest, the *conditional quantile comparator*.

**Definition 1** (Conditional quantile comparator (CQC))**.** *For our triple $(Y, X, A)$, the conditional quantile comparator is the measurable function $g^* : \mathcal{Y} \times \mathcal{X} \to \mathcal{Y}$ such that, for all $y \in \mathcal{Y}$, $\boldsymbol{x} \in \mathcal{X}$,*

$$F_1(g^*(y|\boldsymbol{x})|\boldsymbol{x}) = F_0(y|\boldsymbol{x}).$$

We then simply define $y_1$ as $g^*(y_0|\boldsymbol{x})$. The name conditional quantile comparator derives from the fact that it returns the value of $y_1$ in the equivalent quantile of $Y|X = \boldsymbol{x}, A = 1$ as the quantile of $Y|X = \boldsymbol{x}, A = 0$ that $y_0$ is in.

**Remark 2.** *For simplicity and to ensure such a function is well defined, we will assume that $Y|X = \boldsymbol{x}, A = a$ is a continuous RV for any given $\boldsymbol{x} \in \mathcal{X}$, $a \in \{0, 1\}$ with strictly positive density on its support. We will however allow the support of $Y|X = x, A = a$ to vary in both $\boldsymbol{x}$ and $a$.*

We now introduce another estimand which will serve as a useful stepping stone in our estimation.

**Definition 2** (CCDF contrasting function)**.** *The* CCDF contrasting function *is defined to be* $h^* : \mathcal{Y} \times \mathcal{Y} \times \mathcal{X} \to [-1, 1]$ *given by*

$$h^*(y_0, y_1|\boldsymbol{x}) := F_1(y_1|\boldsymbol{x}) - F_0(y_0|\boldsymbol{x}). \tag{3}$$

This estimand allows following alternative definitions for $g^*$ which help its interpretation and estimation (detailed later). We take $h^{*-1}$ representing the inverse of $h^*$ with respect to the 2$^{\text{nd}}$ argument.

$$g^*(y_0|\boldsymbol{x}) = h^{*-1}(y_0, 0|\boldsymbol{x}) = F_1^{-1}(F_0(y_0|\boldsymbol{x})|\boldsymbol{x}). \tag{4}$$

The second equality still holds if we replace the inverses in the above with generalised inverses. The equality (4) falls straight from the definition of each object and shows how we can use $h^*$ to estimate $g^*$. Moreover, the equality (4) allows us to generalise $g^*$ to discontinuous $Y$ (or pdfs with non-trivial 0 density regions inside the support).

## 2.2 Exploring the CQC

We specifically focus on the CQC, $g^*$, as we feel it gives insightful information allowing comparison of the two distributions $(Y|X, A = 1)$ and $(Y|X, A = 0)$.

The CQC allows us to compare the two distributions beyond simply a single point estimate such as that given by the CATE. This is especially valuable in cases where the two distributions differ beyond just a shift. For example, the effect of some treatments varies greatly between individuals with the same or similar covariates. An example of this is antidepressants, where some patients respond positively while others may have adverse reactions leading to a worse outcome than no treatment whatsoever. Another example is the use of opioids as painkillers where some patients have an increased tolerance making them less effective [12, 22].

As well as being of interest on its own, the quantile comparator relates closely to other estimands of interest. For example, if we take $\Delta^*(y_0|\boldsymbol{x}) := g^*(y_0|\boldsymbol{x}) - y_0$ then $\Delta^*$ tells us whether the equivalent quantile in the treated distribution is higher or lower. This estimand then serves as a heuristic for whether the treatment is beneficial at that untreated response value.

Furthermore, CQTE can be written as

$$\tau_q(\alpha|\boldsymbol{x}) = \Delta^* \left( F_{Y|X, A=0}^{-1}(\alpha|\boldsymbol{x}) \Big| \boldsymbol{x} \right) = g^* \left( F_{Y|X, A=0}^{-1}(\alpha|\boldsymbol{x}) \Big| \boldsymbol{x} \right) - F_{Y|X, A=0}^{-1}(\alpha|\boldsymbol{x}), \tag{5}$$

linking the quantile comparator back to the CQTE. This equivalence highlights the perspective that the CQC can be seen as rephrasing the input of the CQTE in terms of the outcome space.

A key idea within CATE literature is the notion that the CATE itself may be a simpler estimand to study than the marginal treatment outcomes ($\mathbb{E}[Y|X = \boldsymbol{x}, A = a]$) may be individually. One can exploit this feature to improve the CATE's estimation. A similar concept exists with the CQC as we will see in the example below.

**Example 1** (Illustrative Example)**.** *Suppose that*

$$Y|X = x, A = 0 \sim \mathrm{N}(\sin(10x),\ 1^2), \qquad Y|X = x, A = 1 \sim \mathrm{N}(2\sin(10x),\ 2^2).$$

*Then we have $g^*(y|x) = 2y$ which does not depend on $x$ and does not include the sine term present in the individual CDFs. Interestingly, $\mathbb{E}[Y|X, A = 1] - \mathbb{E}[Y|X, A = 0] = \sin(x)$ hence the CATE is still non-constant in this case (the same also holds for the CQTE). Additionally, we have $\Delta^*(y|x) = g^*(y|x) - y = y$ suggesting the intervention is beneficial for positive $y$ and detrimental for negative $y$. We now show 3D plots of a CCDF, the CQC, and the CQTE in Figure 2.*

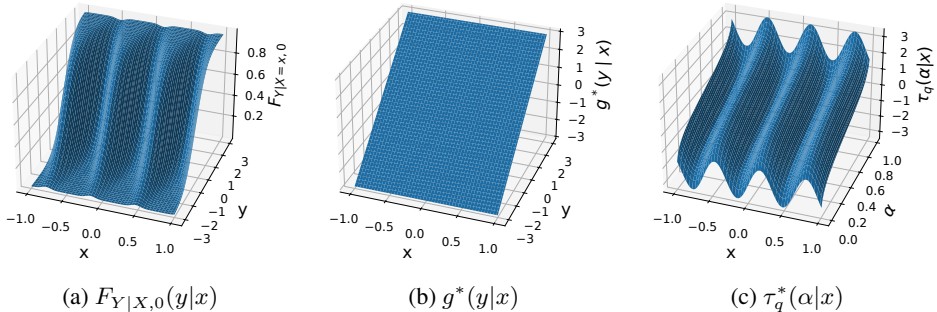

(a) $F_{Y|X,0}(y|x)$        (b) $g^*(y|x)$        (c) $\tau_q^*(\alpha|x)$

Figure 2: Surface plots for CCDF (panel (a)), CQC (panel (b)) and CQTE (panel (c)). We can see that CCDF, and CQTE have high-frequency change in $x$ while the CQC does not depend on $x$.

**Example 2** (General Smoothness Case). *Suppose we are in the potential outcomes framework so that $Y_0, Y_1$ exist with $Y \equiv Y_A$. Now also suppose that $Y_1 = \phi(Y_0, X)$ for some transformation $\phi$ increasing in $Y_0$ for each $X$. Then $\phi$ gives the CQC (i.e. $\phi = g^*$) meaning that smoothness of the CQC can be seen as smoothness of $\phi$. This gives a generalisation of the CATE case where smoothness is present when $Y_1 = Y_0 + \psi(X)$ with $\psi$ smooth*

*A specific example could be a treatment which halves all individuals blood pressure. In this case $\phi^*(y, \boldsymbol{x}) = g^*(y|\boldsymbol{x}) = \frac{1}{2}y$ and so the CQC is smooth but the CQTE and CATE would not be if the individual responses are non-smooth.*

## 3 Estimation procedure

We now describe our estimation procedure for the CQC which is motivated by equation (4). At a high level our approach for estimating $g^*(y_0|\boldsymbol{x})$ will be the following:

- Estimate $h^*(y_0, .|\boldsymbol{x})$ using a pseudo-outcome – a specified proxy response computed from $(Y, X, A)$ which we will regress against.

- Find the value $\hat{y}_1$ which makes our estimate of $h^*(y_0, .|\boldsymbol{x})$ closest to 0.

Section 3.1 and Algorithm 1 give our $h^*$ estimation procedure while Section 3.3 and Algorithm 2 give our $g^*$ estimation procedure.

### 3.1 Estimating the CCDF contrasting function $h^*$

We focus on estimating $h^*$ primarily for its two nice properties:

1. Similar to $g^*$, $h^*$ can exhibit smoothness even when the individual CCDFs are not smooth.

2. The estimation of $h^*$ can be re-framed as a CATE problem.

The first property is important as the smoothness of $h^*$ determines the best estimation rate that can be achieved when using non-parametric regression, setting a target for our approach. In particular, smoother functions have better estimation rates. The second property is important as it gives us a method for attaining this target rate. By re-framing the estimation as a CATE problem, we can leverage existing results to build a robust estimator which achieves the target estimation accuracy rate even when the rate of estimating nuisance parameters is sub-optimal. We demonstrate this robustness later using finite sample bounds on the estimation accuracy (Proposition 1 & Theorem 2).

First, we show how the estimation of $h^*$ can be solved using a CATE estimator. Note that

$$h^*(y_0, y_1|\boldsymbol{x}) = \mathbb{E}[\mathbb{1}\{Y \leq y_1\}|X = \boldsymbol{x}, A = 1] - \mathbb{E}[\mathbb{1}\{Y \leq y_0\}|X = \boldsymbol{x}, A = 0].$$

Hence, for a given $y_0, y_1$, if we define the RV $W_{y_0, y_1} := \mathbb{1}\{Y \leq y_A\}$, then estimating $h^*(y_0, y_1|.)$ is equivalent to estimating the CATE with $W_{y_0, y_1}$ replacing $Y$ as the response. To perform this estimation, we turn to a recent method developed by Kennedy [15]. They propose to write the CATE as a conditional expectation of a function of $Z$ called a pseudo-outcome. A robust estimator

is then obtained by regressing this pseudo-outcome against $X$. In our setting, the pseudo-outcome with the new response $W_{y_0,y_1}$, for a sample $(y, \boldsymbol{x}, a)$ is given by

$$\varphi_{y_0,y_1}(y, \boldsymbol{x}, a) := \frac{a - \pi(\boldsymbol{x})}{\pi(\boldsymbol{x})(1 - \pi(\boldsymbol{x}))} \{\mathbb{1}\{y \le y_a\} - F_a(y_a|\boldsymbol{x})\} + F_1(y_1|\boldsymbol{x}) - F_0(y_0|\boldsymbol{x}) \quad (6)$$

$$= \frac{a - \pi(\boldsymbol{x})}{\pi(\boldsymbol{x})(1 - \pi(\boldsymbol{x}))} \{\mathbb{1}\{y \le y_a\} - F_a(y_a|\boldsymbol{x})\} + h^*(y_0, y_1|\boldsymbol{x}).$$

Since $h^*(y_0, y_1|\boldsymbol{x}) = \mathbb{E}[\varphi_{y_0,y_1}(Z)|X = \boldsymbol{x}]$ (Proposition 5, Appendix C.1), regressing $\varphi_{y_0,y_1}(Z)$ on $X$ provides an estimate for $h^*$.

As we do not know the CDFs nor the propensity score, we need to replace them in (6) with estimates. We define $\hat{\pi}, \hat{F}_0, \hat{F}_1$ to be estimates of $\pi, F_0, F_1$ respectively. We then construct $\hat{\varphi}_{y_0,y_1}$ in the same way as $\varphi_{y_0,y_1}$, but using estimated quantities $\hat{\pi}$, $\hat{F}_a$ instead. $\hat{\varphi}_{y_0,y_1}$ can now serve as the pseudo-outcome in our regression. We also use sample splitting to de-correlate the propensity score and CDF estimates from the $h^*$ estimate. This helps make our estimator doubly robust, as we will see in the following theory.

We are now ready to define our Doubly Robust (DR)-learner to estimate $h^*$ in Algorithm 1.

---

**Algorithm 1** DR estimation procedure for the CCDF contrasting function $h^*$

---

**Require:** $y_0, y_1 \in \mathcal{Y}$, Data $D$, a regressor (e.g. linear smoother)
 1: Define $\mathcal{I} := [n]$, $\mathcal{J} := \{n+1, \dots, 2n\}$ and split $D$ into $D_{\mathcal{I}} := \{Z_i\}_{i \in \mathcal{I}}, D_{\mathcal{J}} := \{Z_j\}_{j \in \mathcal{J}}$.
 2: Using $D_{\mathcal{I}}$ to estimate $\hat{\pi}, \hat{F}_0(y_0|.), \hat{F}_1(y_1|.)$ by regressing $\mathbb{1}\{A = 1\}, \mathbb{1}\{Y \le y_0\}, \mathbb{1}\{Y \le y_1\}$ respectively against $X$.
 3: Use these estimates to obtain $\hat{\varphi}(Z_j)$ for $j \in D_{\mathcal{J}}$ using equation (6)
 4: Using $D_{\mathcal{J}}$ to regress $\hat{\varphi}$ against $X$ to obtain estimate $\hat{h}(y_0, y_1|.)$ of $h^*(y_0, y_1|.)$.

---

**Remark 3.** *Cross-fitting can be implemented by repeating the procedure with the roles of $\mathcal{I}$ and $\mathcal{J}$ switched and then averaging the two estimates of $h^*$. We could also perform this procedure multiple times with different random splits of the data to improve our estimator's potential stability.*

Note that our algorithm is not specific on which form of regression to use allowing for any parametric or non-parametric procedure. Further to this, it can also be easily adapted to use other pseudo-outcome procedures such as the R-learner of Nie and Wager [23] or a standard inverse propensity weighting approach which we describe in Appendix A.3.

### 3.2 Finite sample bound of $h^*$ estimator

In this section, we prove the estimation accuracy of $\hat{h}$, which will play important roles in the following notation and assumptions we need for estimating $g^*$. These accuracy statements will be made for an arbitrarily fixed $\boldsymbol{x} \in \mathcal{X}$. For our theoretical and experimental results we use linear smoothers for the final regression. This means our estimate $\hat{h}$ and oracle estimate $h'$ are of the form

$$\hat{h}(y_0, y_1|\boldsymbol{x}) = \sum_{j \in \mathcal{J}} w_j \hat{\varphi}_{y_0,y_1}(Z_j) \qquad h'(y_0, y_1|\boldsymbol{x}) = \sum_{j \in \mathcal{J}} w_j \varphi_{y_0,y_1}(Z_j)$$

where the weights $w_j \equiv w_j(\boldsymbol{x}, X_{\mathcal{J}})$ are constructed using $X_{\mathcal{J}} := \{X_j\}_{j \in \mathcal{J}}$ with $w_j \ge 0$ and $\|\boldsymbol{w}\|_1$. Linear smoothers encompass a broad class of estimation techniques used in both low and high-dimensional settings. Examples include k-NN regression [8, 9], kernel ridge regression [29], generalised forests [3], and Mondrian forests [18]. Additionally linear smoothers have been shown to adapt to intrinsic low dimensionality in regression problems in higher dimensions [16], making them an apt estimator for our purposes.

For $\phi : \mathcal{Z} \to \mathbb{R}$ treated as deterministic and $p > 1$, we also define the norms

$$\|\phi\|_{\boldsymbol{w}^s} := \sqrt{\frac{1}{\|\boldsymbol{w}^s\|_1} \sum_{i \in \mathcal{J}} w_j^s \, \mathbb{E}\left[\phi(Z)^2 | X = X_i\right]}$$

and take $\|\phi\|_{\boldsymbol{w}} := \|\phi\|_{\boldsymbol{w}^1}$. Note that this norm is random as the weights depend upon $X_{\mathcal{J}}$.

We now aim to show that we are able to exploit smoothness in $h^*$ even when the CCDFs and propensity score are less smooth. We introduce the notion of smoothness through Hölder functions.

**Definition 3** (Hölder functions). *We say that a function $f : \mathcal{X} \to \mathbb{R}$ is $(\gamma, C)$-Hölder for $\gamma \in (0, 1], C \geq 1$ if for any $\boldsymbol{x}', \boldsymbol{x}'' \in \mathcal{X}$,*

$$|f(\boldsymbol{x}') - f(\boldsymbol{x}'')| \leq C\|\boldsymbol{x}' - \boldsymbol{x}''\|^\gamma.$$

Here, larger $\gamma$ represents a smoother function which can be estimated at faster rates.

**Assumption 1.** *For any $y_0, y_1 \in \mathcal{Y}, \; \delta > 0, a \in \{0, 1\}$:*

*(a) There exists $\xi \in (0, 1/2]$ such that $\pi(\boldsymbol{x}), \hat{\pi}(\boldsymbol{x}') \in [\xi, 1 - \xi]$ for all $\boldsymbol{x}' \in \mathcal{X}$.*

*(b) With probability at least $1 - \delta$, $\|\hat{\varphi}_{y_0, y_1} - \varphi_{y_0, y_1}\|_{\boldsymbol{w}^2} \leq \varepsilon_{\hat{\varphi}}(n, \delta)$.*

*(c) With probability at least $1 - \delta$, $\|\hat{\pi} - \pi\|_{\boldsymbol{w}} \leq \varepsilon_\alpha(n, \delta)$ and $\|F_a(y_a|.) - \hat{F}_a(y_a|.)\|_{\boldsymbol{w}} \leq \varepsilon_\beta(n, \delta)$.*

*(d) For $\gamma \in (0, 1], \; C \geq 1, \; h^*(y_0, y_1|.)$ is $(\gamma, C)$-Hölder.*

Assumption 1(a) exists to ensure for any covariate value, neither treatment assignment has too low a probability. Assumption 1(b) controls the convergence of the estimated pseudo-outcome to the true pseudo-outcome. Assumption 1(c) sets up the smoothness of the propensity score and CCDFs alongside the convergence rates of their estimators as $\varepsilon_\alpha, \varepsilon_\beta$ respectively. Assumption 1(d) sets up the smoothness of $h^*$ which will control the convergence rate of the oracle estimation procedure. Assumptions 1 (c) and (d) control the accuracy of our estimator in the following result.

**Proposition 1.** *Suppose that Assumption 1 holds and let $\hat{h}$ be a linear smoother estimated as in Algorithm 1. Then for any $y_1, y_0 \in \mathcal{Y}, \; \delta \in (0, 2/e]$ and our $\boldsymbol{x} \in \mathcal{X}$, with probability at least $1 - \delta$,*

$$\left| \hat{h}(y_0, y_1|\boldsymbol{x}) - h^*(y_0, y_1|\boldsymbol{x}) \right| \leq \varepsilon_h(n, \delta).$$

*Here, for each $\delta \in (0, 2/e]$ we have*

$$\varepsilon_h(n, \delta) := \sqrt{2\log(8/\delta)/n}\,\varepsilon_{\hat{\varphi}}(n, \delta/4) + \varepsilon_\alpha(n, \delta/4)\varepsilon_\beta(n, \delta/4) + \varepsilon_\gamma(n, \delta/4),$$

$$\varepsilon_\gamma(n, \delta) := |\mathbb{E}[h'(y_0, y_1|\boldsymbol{x}) - h^*(y_0, y_1|\boldsymbol{x})|X_{\mathcal{J}}, D_{\mathcal{I}}]|$$
$$+ \sqrt{2\log(2/\delta)/n}\,\|\boldsymbol{w}\|\|\varphi - h(y_0, y_1|.)\|_{\boldsymbol{w}^2} + 2\|\boldsymbol{w}\|_\infty \log(2/\delta)/(3\xi).$$

We have that $\varepsilon_\gamma$ gives an upper bound on the accuracy of the oracle estimation and so acts as a target for our estimation procedure. If $\varepsilon_{\hat{\varphi}}(n, \delta) \to 0$ then the first term in $\varepsilon_h$ is guaranteed to be $o(\varepsilon_\gamma(n, \delta))$ for fixed $\delta$. Hence the first and last terms converge at oracle rates with respect to $n$. As the $\varepsilon_\alpha$ and $\varepsilon_\beta$ terms are multiplied together we can obtain better rates than either of them individually have. This is because both $\varepsilon_\alpha$ and $\varepsilon_\beta$ can converge to 0 slower than $\varepsilon_\gamma$ while their product $\varepsilon_\alpha \cdot \varepsilon_\beta$ converges quicker. This provides the desired double robustness as our estimation can converge at oracle rates even when convergence for the nuisance parameters is slower. Now that we have this we can convert our estimate of $h^*$, into an estimate of $g^*$.

### 3.3 Estimating the conditional quantile comparator $g^*$

In order to obtain an estimate of $g^*(y_0|\boldsymbol{x})$ at a fixed $y_0, \boldsymbol{x}$, we need to obtain estimates of $h^*(y_0, y_1|\boldsymbol{x})$ at various values of $y_1$. As $h^*$ is monotonic, we would then like to search for a monotonic function which aligns with these estimates. This monotonicity is especially important because it allows us to bound the estimation accuracy of $\hat{h}$ uniformly over all $y_1$ at a similar rate to our pointwise accuracy. With this, we can easily translate the estimation accuracy in $\hat{h}$ (obtained in proposition 1) into the accuracy in $\hat{g}$. Additionally, it simplifies the process of inverting $\hat{h}$. In general our method in Algorithm 1 will not produce monotonic $\hat{h}$ (this is in contrast to some other approaches such as an IPW pseudo-outcome, see Appendix A.3), or separately estimating the CCDFs). We can however obtain a monotonic estimate of $h^*$ using isotonic projection.

**Definition 4** (Isotonic Projection). *We define the isotonic projection of $\boldsymbol{\alpha}' \in \mathbb{R}^p$ as follows:*

$$P(\boldsymbol{\alpha}') := \underset{\boldsymbol{\alpha} \in \text{Iso}(p)}{\operatorname{argmin}} \|\boldsymbol{\alpha} - \boldsymbol{\alpha}'\|$$

*where $\text{Iso}(p) := \{\boldsymbol{\alpha} \in \mathbb{R}^p | \alpha_j \leq \alpha_{l+1} \; \forall l \in [p - 1]\}$, the set of all isotonic vectors in $\mathbb{R}^p$.*

**Remark 4.** *We can use the Pool Adjacent Violators Algorithm (PAVA) [6] which performs isotonic projection and is implemented in the* `IsotonicRegression` *class of sci-kit learn in Python [25].*

Hence, for a fixed value of $(y_0, \boldsymbol{x})$ and a set of predictions $\hat{\alpha}_l = \hat{h}(y_0, y_1^{(l)}|\boldsymbol{x})$ with $y^{(l)} \leq y^{(l+1)}$, we can take $\tilde{\boldsymbol{\alpha}} := P_{\text{Iso}(p)}(\hat{\boldsymbol{\alpha}})$ and use these to obtain a new monotonic estimate of $h^*$. Furthermore, by a result in Yang and Barber [32], $\tilde{\boldsymbol{\alpha}}$ will be at least as accurate as $\hat{\boldsymbol{\alpha}}$ in the worst case. We now describe our approach for estimating the CQC using this projection approach in Algorithm 2.

---

**Algorithm 2** DR estimation procedure for the CQC $g^*$

---

**Require:** Data $D$; test point $(y_0, \boldsymbol{x})$; sorted evaluation points $\{y^{(l)}\}_{l=1}^p$
  1: Apply Algorithm 1 to obtain estimate of $h^*$ given by $\hat{h}$.
  2: Define $\hat{\alpha}_l := \hat{h}(y_0, y^{(l)}|\boldsymbol{x})$ for $l \in [p]$.
  3: Isotonically project $\hat{\boldsymbol{\alpha}}$ using PAVA to obtain $\tilde{\boldsymbol{\alpha}}$ with $\tilde{\alpha}_i \leq \tilde{\alpha}_{i+1}$.
  4: Take $\hat{g}(y_0|\boldsymbol{x}) := y^{(l^*)}$ with $l^* := \text{argmin}_{l \in [p]}|\tilde{\alpha}_l|$.

---

**Remark 5.** *For the case where $\hat{h}$ is a step function, these steps can serve as our evaluation points while for continuous $\hat{h}$ one could take these candidate $y_1$ points at small evenly spaced intervals. Empirically we also find that $\hat{h}$ is already close to isotonic and so step 3. of the algorithm is mostly for the theoretical justification of our approach.*

While it may seem inefficient to be estimating the CQC via the CCDF contrasting function and then inverting, due to the monotonicity of $h^*$, we actually pay a very small cost in estimation accuracy for having to estimate the CCDF contrasting function over all $y_1$. We make this notion more explicit in the following section.

### 3.4 Finite sample bound of the CQC estimator

We now provide the accuracy of our estimate $\hat{g}$ obtained by Algorithm 2 when used in conjunction with linear smoothers. We assume that $\hat{F}_a$ are also fit using linear smoothers of the form

$$\hat{F}_a(y|\boldsymbol{x}', a) = \sum_{i \in \mathcal{I}} w_{F_a;i}(\boldsymbol{x}'; X_\mathcal{I}, A_\mathcal{I})\mathbb{1}\{Y_i \leq y\}$$

with $w_j(\boldsymbol{x}') \equiv w_j(\boldsymbol{x}', X_\mathcal{I}, A_\mathcal{I}) > 0, \|\boldsymbol{w}(\boldsymbol{x}')\|_1 = 1$. We will also require the following assumptions.

**Assumption 2.** *For our RV $X$, any $y \in \mathcal{Y}$, $\boldsymbol{x}' \in \mathcal{X}$, $\delta < e^{-1}$:*

*(a) There exists some $s, \eta > 0$ such that $F_1(y'|\boldsymbol{x}) \geq \eta$ for all $y' \in B_s(g^*(y|\boldsymbol{x}))$.*

*(b) W.p. at least $1 - \delta$, $\max_{j \in \mathcal{J}} w_j \leq \varepsilon_{\boldsymbol{w}}(n, \delta)$ and $\max_{i \in \mathcal{I}} w_{F_a;i}(X) \leq \varepsilon_{\boldsymbol{w}}(n, \delta)$.*

Assumption 2(a) is a mild assumption which allows us to convert the $\hat{h}$ accuracy into $\hat{g}$ accuracy while Assumption 2(b) bounds the rates of decay of the weights in our linear smoothers.

**Theorem 2.** *Let $\hat{g}$ be estimated as using Algorithm 2 with $\{Y_i\}_{i \in I_1}$, sorted and then used as our evaluation points and linear smoothers used for regressions in Algorithm 1. Then provided Assumptions 1 & 2 hold we have that for $\delta \in (0, e^{-1})$ and sufficiently large $n$, w.p. at least $1 - \delta$,*

$$|\hat{g}(y|\boldsymbol{x}) - g^*(y|\boldsymbol{x})| \leq 2\left(\eta^{-1}\varepsilon_h(n, \delta/(2n)) + \xi^{-1}\varepsilon_{\boldsymbol{w}}(n, \delta/(2n))\right).$$

From this result we see that if our weights decay at rate faster than $\varepsilon_h(n, \delta)$ then this error will be dominated by the $\varepsilon_h$ term. We believe this to hold in most cases and show that it does comfortably when using Nadaraya-Watson (NW) estimation [21, 31] with a box kernel in Appendix C.4. Furthermore, if the dependence on $\delta$ in both terms is of the form $\log^c(1/\delta)$ for some $c > 0$ then we obtain the same rate of estimation as for $h^*$ up to polylog factors. This means we translate our desirable double robustness $\hat{h}$ over to $\hat{g}$. We also obtain finite sample bounds on $\mathbb{E}[|\hat{g}(Y|\boldsymbol{x}) - g^*(Y|\boldsymbol{x})| \mid A = 0, \hat{g}]$ with high probability and present this in Appendix C.3.

## 4 Numerical experiments

We now apply our approach to a series of simulated and real data scenarios in order to demonstrate the utility of our estimand and the effectiveness of our estimation procedure. For these, we use NW estimation as our regression procedure throughout. See appendix A.2 for details on NW estimation.

## 4.1 Simulated experiment

In this section, we test our method's performance in terms of our estimator's mean absolute error under simulated scenarios.[1] In each scenario we test against a separate estimator which estimates the two CCDFs separately and simply takes their difference, an IPW pseudo-outcome estimator detailed in Appendix A.3, the CQTE estimator of Kallus and Oprescu [14], and the oracle DR estimator where $\hat{\varphi}$ is replaced with $\varphi$ (i.e. exact $\pi, F_a$ are used). In this experiment, we return back to the set-up of example 1. We now change the frequency of the sine term by taking

$$Y|X = x, A = 0 \sim \mathrm{N}(\sin(\gamma\pi x), \ 1^2), \qquad Y|X = x, A = 1 \sim \mathrm{N}(2\sin(\gamma\pi x), \ 2^2),$$
$$\pi(x) = 0.4\sin(\gamma\pi x) + 0.5.$$

for $\gamma \in [0, 10]$ so that increasing $\gamma$ imitates decreasing smoothness of our nuisance parameters. In our experiments half the samples are used to estimate the propensity score and CCDFs and the other half are used to regress against the pseudo-outcome. Our estimate $\hat{g}$ is then compared against $g^*$ using a hold-out testing set. This process is repeated 500 times with new training data on each run. From this, a Monte-Carlo estimate of $\mathbb{E}_{\hat{g}}[\mathbb{E}_X[\mathbb{E}_{Y|X,A=0}[|\hat{g}(Y|X) - g^*(Y|X)|]]]$ is produced alongside 95% confidence intervals (CIs). In our first experiment, we let $2n = 1000$ and vary $\gamma$ in $[0, 10]$. In our second experiment, we let $\gamma = 6$ and $2n$ vary in $[200, 5000]$. The results of this are shown in Figure 3.

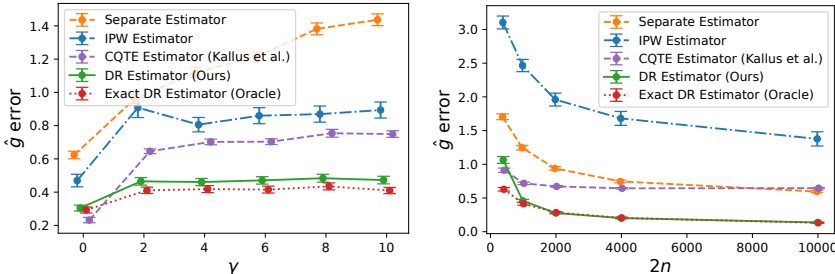

Figure 3: Mean absolute error with 95% CIs for various estimators. The left plot has fixed sample size ($2n = 1000$) and increasing $\gamma$. The right plot has $\gamma = 6$ and increasing sample size.

We can see that the average error decreases as the sample size $2n$ grows and mildly increases as $\gamma$ grows. This is expected as increasing $\gamma$ makes estimating the nuisance parameters more challenging. The result shows that the proposed DR method achieves the best performance compared with the Separate and IPW estimators and is only marginally outperformed by the oracle estimator. Additionally we see that the method of Kallus and Oprescu [14] is much more affected by the increase in $\gamma$. This is because unlike the CQC, the CQTE has a complexity that depends on the frequency term ($\gamma$). We also observe much better performance as the sample size increases. We hypothesise that the plateau in the CQTE approach is due to the difficulty of estimating the reciprocal of the PDF, which causes the estimation to be unstable irrespective of sample size. Further simulated experiments including 10-dimensional $x$ and linear CQC are given in Appendix B.1.

## 4.2 Real world employment example

To show the performance in the real-world scenarios, we use a dataset on an employment programme which has been studied in various prior works [4, 5, 26]. Within the programme, some participants were given job placements or temporary help jobs while others received no intervention. Participants' earnings were then monitored over the next 8 quarters following their enrolments. We take their net earnings as our response ($Y$) and the employment intervention as the treatment ($A = 1$). We use each participant's age at their entry to the study as our covariate ($X$). We fit our quantile comparator function on 2,000 participants. Figure 4 shows our estimate of $\Delta^*(y|\boldsymbol{x}) = g^*(y|\boldsymbol{x}) - y$ for various values of $(y, \boldsymbol{x})$.

We see that participants around age 23 and between ages 32-37 benefit most from this scheme, as indicated by the darker colour on the heat map. Additionally, the lower quantile of the income distribution (wage $\leq \$7500$) shows the least change, indicating that wage improvements primarily occur

---

[1] Code implementation can be found at: github.com/joshgivens/ConditionalOutcomeEquivalence

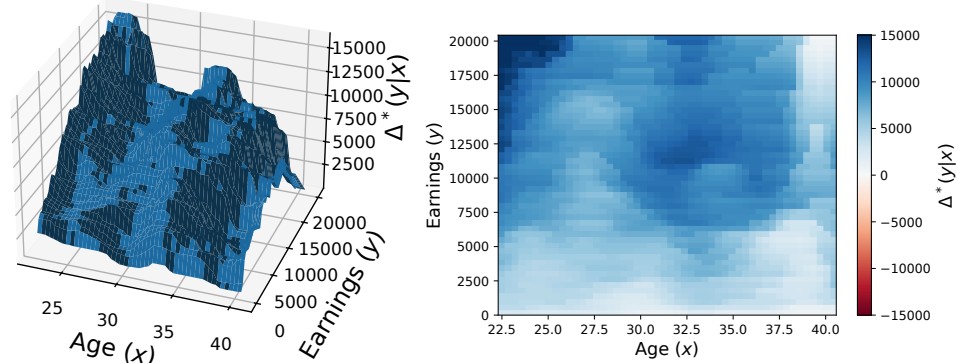

Figure 4: Surface and heat plot of $\Delta^*(y|\boldsymbol{x})$ for our employment data with $X$ =Age, $Y$=Income.

for the higher income group. For participants aged 40, there appears to be little change in outcomes overall. To demonstrate our approach in a medical setting, we apply our method to evaluate the effectiveness of a colon cancer treatment, as detailed in Appendix B.3. For comparative purposes, we also provide an estimate of the CQTE for this data in Appendix B.2.

## 5   Limitations

The theory provided here gives a strong foundation and motivation for framing our problem in this particular manner. There is however a great deal more to be explored in this area from a theoretical perspective. For example, one immediate improvement would be to give a more general case where our weight decay (in Assumption 2 (b) for Theorem 2) is sufficiently fast. In addition more work needs to be done exploring the relationship between the smoothness of $g^*$ and the smoothness $h^*$. Smoothness in $h^*$ appears to be a stronger condition so ideally we would like to make theoretical statements directly on the smoothness of $g^*$. Interestingly our experiments on synthetic data do seem to suggest that it is the complexity of $g^*$ which drives the estimation rate as in these experiments $h^*$ increases in complexity while $g^*$ remains constant. Additionally, changing our experiment in Section 4.1 to have uniform response so that both $h^*, g^*$ are constant rather than just $g^*$, seems to give no material improvement to the performance of our estimator (see Appendix B.1.3).

Another limitation of our current estimation procedure is that it requires learning an estimand and then inverting it to obtain our final estimator. While this process is relatively simple, it could be streamlined and made more computationally efficient if we could produce a more direct estimator similar to the DR-Learner for CATE [15] or the CQTE estimator in Kallus and Oprescu [14].

Finally, while we have been able to provide more concrete examples of smoothness for the CQC these are still limited to the case of a deterministic treatment effect which we would like to expand upon. This is closely related to a more general limitation with quantile-based estimands, the CQC included, in that they lack meaningful interpretability for the individual. While the CATE can be viewed as the expected difference in an individual's outcome on and off the treatment, no such individual-level interpretation exists for the CQC or indeed any other estimand trying to learn higher level distributional information than the mean. As a result the CATE is still a more naturally interpretable estimand. To facilitate this interpretation however, one still needs to make assumptions about a lack of confounding between the treatment assignment and the potential outcomes, which are only verifiable in certain restrictive scenarios [30].

## 6   Conclusion

In this paper we have introduced a new treatment effect estimand, the conditional quantile comparator and demonstrated its efficacy both in terms of its doubly robust estimation, and its ability to provide valuable data insights. This is a promising direction as it allows quantile-based treatment effect exploration to "keep up" with the CATE in terms of estimation quality offering more flexibility as to which estimand can be used to best describe the data. For these reasons, we see the CQC as an exciting and worthwhile new direction within the HTE framework.

## Acknowledgments and Disclosure of Funding

Josh Givens was supported by a PhD studentship from the EPSRC Centre for Doctoral Training in Computational Statistics and Data Science (COMPASS).

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

# A    Additional details

## A.1    Notation table

Table 1: Table of notation.

| Notation | Definition | Description |
|---|---|---|
| $Y$ | RV on $\mathcal{Y} \subseteq \mathbb{R}$ | Response |
| $X$ | RV on $\mathcal{X} \subseteq \mathbb{R}^d$ | Covariates |
| $A$ | RV on $\{0,1\}$ | Treatment |
| $Z$ | $(Y, X, A)$ | |
| $\pi(\boldsymbol{x})$ | $\mathbb{P}(A=1\|X=\boldsymbol{x})$ | Propensity Score |
| $F_a(y\|\boldsymbol{x})$ | $\mathbb{P}(Y \leq y\|X=\boldsymbol{x}, A=a)$ | Conditional CDF (CCDF) |
| $F_a^{-1}(\alpha\|\boldsymbol{x})$ | $\inf \{y \in \mathcal{Y}\|F_a(y\|\boldsymbol{x}) \geq \alpha\}$ | Conditional quantile function |
| $\tau(\boldsymbol{x})$ | $\mathbb{E}[Y\|X=\boldsymbol{x}, A=1] - \mathbb{E}[Y\|X=\boldsymbol{x}, A=0]$ | Conditional average treatment effect (CATE) |
| $\tau_q(\alpha\|\boldsymbol{x})$ | $F_1^{-1}(\alpha\|\boldsymbol{x}) - F_0^{-1}(\alpha\|\boldsymbol{x})$ | Conditional quantile treatment effect (CQTE) |
| $D$ | $\{Z_i\}_{i=1}^{2n}$ | IID data sample |
| $I_a$ | $\{i \in [2n]\|A_i = 1\}$ | Treatment $a$ index |
| $D_a$ | $\{Z_i\}_{i \in I_a}$ | |
| $n_a$ | $\|I_a\|$ | |
| $[n]$ | $\{1, \ldots, n\}$ | |
| $\|\boldsymbol{x}\|$ | $\sqrt{\frac{1}{d} \sum_{i=1}^d x_i^2}$ | Euclidean Norm |
| $\|\boldsymbol{x}\|_1$ | $\frac{1}{d} \sum_{j=1}^d \|x_j\|$ | 1-norm |
| $\|\boldsymbol{x}\|_\infty$ | $\max_{j \in [d]} \|w_j\|$ | $\infty$-norm |
| $g^*(y\|\boldsymbol{x})$ | $F_1^{-1}(F_0(y\|\boldsymbol{x})\|\boldsymbol{x})$ | Conditional quantile comparator CQC |
| $h^*(y_0, y_1\|\boldsymbol{x})$ | $F_1(y_1\|\boldsymbol{x}) - F_{Y\|X,0}(y_0\|\boldsymbol{x})$ | CCDF contrasting function |
| $\Delta^*(y\|\boldsymbol{x})$ | $g^*(y\|\boldsymbol{x}) - y$ | Quantile difference function |
| $\hat{f}$ | Estimate of $f/f^*$ | |
| $\varphi_{y_0, y_1}(y, \boldsymbol{x}, a)$ | $\frac{a-\pi(\boldsymbol{x})}{\pi(\boldsymbol{x})(1-\pi(\boldsymbol{x}))} \{\mathbb{1}\{y \leq y_a\} - F_a(y_a\|\boldsymbol{x})\}$ $+ F_1(y_1\|\boldsymbol{x}) - F_0(y_0\|\boldsymbol{x})$ | The Pseudo-Outcome |
| $\hat{\varphi}_{y_0, y_1}(y, \boldsymbol{x}, a)$ | $\varphi$ with $\pi, F_a$ replaced by $\hat{\pi}, \hat{F}_a$ | |
| $\mathcal{I}, \mathcal{J}$ | $\{1, \ldots, n\}, \{n+1, \ldots, 2n\}$ | Sample splitting indices |
| $D_{\mathcal{I}}, D_{\mathcal{J}}$ | $\{Z_i\}_{i \in \mathcal{I}}, \{Z_j\}_{j \in \mathcal{J}}$ | Data split |
| $\boldsymbol{w} \in \mathbb{R}^n$ | $w_j \equiv w_j(\boldsymbol{x})$ | $\hat{h}$ estimation weights |
| $\hat{h}(y_0, y_1\|\boldsymbol{x})$ | $\sum_{j \in \mathcal{J}} w_j(\boldsymbol{x}) \hat{\varphi}_{y_0, y_1}(Z_j)$ | $h^*$ estimate |
| $h'(y_0, y_1\|\boldsymbol{x})$ | $\sum_{j \in \mathcal{J}} w_j(\boldsymbol{x}) \varphi_{y_0, y_1}(Z_j)$ | Oracle $h^*$ estimate |
| $\|\phi\|_{\boldsymbol{w}^s}$ | $\sqrt{\|\boldsymbol{w}^s\|_1^{-1} \sum_{i \in \mathcal{J}} w_j^s \mathbb{E}[\phi(Z)^2\|X=X_i]}$ | |
| $\xi$ | $\pi(\boldsymbol{x}') \in [\xi, 1-\xi]$ for all $\boldsymbol{x}' \in \mathcal{X}$ | |
| $\varepsilon_{\hat{\varphi}}(n, \delta)$ | $\|\hat{\varphi} - \varphi\|_{\boldsymbol{w}^2} \leq \varepsilon_{\hat{\varphi}}(n, \delta)$ w.p. $1-\delta$ | $\hat{\varphi}$ error bound w.h.p. |
| $\varepsilon_\alpha(n, \delta)$ | $\|\hat{\pi} - \pi\|_{\boldsymbol{w}^2} \leq \varepsilon_\alpha(n, \delta)$ w.p. $1-\delta$ | $\hat{\pi}$ error bound w.h.p. |
| $\varepsilon_\beta(n, \delta)$ | $\|\hat{F}_a(y\|.) - F_a(y\|.)\|_{\boldsymbol{w}^2} \leq \varepsilon_\beta(n, \delta)$ w.p. $1-\delta$ | $\hat{F}_a$ error bound w.h.p. |
| $\gamma$ | Smoothness of $h^*(y_0, y_1\|.)$ | |
| $\varepsilon_\gamma(n, \delta)$ | $\mathbb{E}[h'(y_0, y_1\|\boldsymbol{x}) - h^*(y_0, y_1\|\boldsymbol{x})\|X_{\mathcal{J}}, D_{\mathcal{I}}]$ $+ \sqrt{2\log(1/\delta)/n} \|\boldsymbol{w}\| \|\varphi - h(y_0, y_1\|.)\|_{\boldsymbol{w}^2}$ $+ 2\xi^{-1}\|\boldsymbol{w}\|_\infty \varepsilon_\delta(n)/3$ | $h'$ error bound w.h.p. |
| $\varepsilon_{T_n}(n, \delta)$ | $\sqrt{2\log(1/\delta)/n} \varepsilon_{\boldsymbol{w}}(n, \delta) \varepsilon_{\hat{\varphi}}(n, \delta)$ | $T_n(\boldsymbol{x})$ error bound w.h.p. |
| $\varepsilon_h(n, \delta)$ | $\varepsilon_{T_n}(n, \delta/4) + (\varepsilon_\alpha \cdot \varepsilon_\beta)(n, \delta/4) +$ $\varepsilon_\gamma(n, \delta/4)$ | $\hat{h}$ error bound w.h.p. |
| $\text{Iso}(p)$ | $\{\boldsymbol{\alpha} \in \mathbb{R}^p\|\alpha_l \leq \alpha_{l+1} \forall l \in [p-1]\}$ | Set of isotonic vectors |
| $P_{\text{Iso}(p)}(\boldsymbol{\alpha}')$ | $\text{argmin}_{\boldsymbol{\alpha} \in \text{Iso}(p)} \|\boldsymbol{\alpha} - \boldsymbol{\alpha}'\|$ | Isotonic projection |
| $\boldsymbol{w}_{F_a} \in \mathbb{R}^n$ | $w_{F_a;i}(\boldsymbol{x}) \equiv w_{F_a;i}(\boldsymbol{x}', X_{\mathcal{I}})$ | $\hat{F}_a$ estimation weights |

| Notation | Definition | Description |
|---|---|---|
| $\hat{F}_a(y\|\boldsymbol{x}')$ | $\sum_{i \in \mathcal{J}} w_{F_a}(\boldsymbol{x})\mathbb{1}\{Y_j \leq y\}$ | CCDF estimate |
| $\eta$ | $F_1(y'\|\boldsymbol{x}) > \eta \, \forall, y' \in B_s(g^*(y\|\boldsymbol{x}))$ | Density lower bound |
| $\varepsilon_{\boldsymbol{w}}(n,\delta)$ | $\max\{\|\boldsymbol{w}(\boldsymbol{x})\|_\infty, \|\boldsymbol{w}_{F_a}(X)\|_\infty\}$ w.p. $1-\delta$ | Variance lower bound w.h.p. |
| | Appendix Notation | |
| $k(\boldsymbol{x},\boldsymbol{x}')$ | | NW estimation kernel |
| $m_f(\boldsymbol{x})$ | $\mathbb{E}[f(Z)\|X = \boldsymbol{x}]$ | |
| $\hat{m}_f(\boldsymbol{x})$ | $\sum_{j \in \mathcal{J}} w_j(\boldsymbol{x})f(Z_j)$ | |
| $\hat{b}(\boldsymbol{x})$ | $m_{\hat{\varphi}-\varphi}(\boldsymbol{x})$ | |
| $\varepsilon_\delta(n)$ | $\max\{\log(1/\delta)/n, 1/n\}$ | |
| $[n]_0$ | $\{0, 1, \dots, n\}$ | |
| $f(\Delta y)$ | $\lim_{\varepsilon \downarrow 0} f(y) - f(y - \varepsilon)$ | Step size of $f$ at $y$ |

## A.2 Nadaraya-Watson estimation

Throughout, we use NW estimation as our standard non-parametric regression technique. For a kernel $k : \mathcal{X} \times \mathcal{X} \to [0, \infty)$, IID data sample $D := \{(Y_i, X_i)\}_{i=1}^n$, and $\boldsymbol{x} \in \mathcal{X}$ the NW estimate of $\mathbb{E}[Y|X = \boldsymbol{x}]$ is given by

$$\sum_{i=1}^n \frac{k(\boldsymbol{x}, X_i)}{\sum_{j=1}^n k(\boldsymbol{x}, X_j)} Y_i.$$

Applying this to our pseudo-outcome regression in Algorithm 1, we get that

$$\hat{h}(y_0, y_1|\boldsymbol{x}) := \sum_{i \in \mathcal{J}} \frac{k(\boldsymbol{x}, X_i)}{\sum_{j \in \mathcal{J}} k(\boldsymbol{x}, X_j)} \hat{\varphi}_{y_0,y_1}(Y_i, X_i, A_i) \tag{7}$$

$$= \sum_{i \in \mathcal{J}} \frac{k(\boldsymbol{x}, X_i)}{\sum_{j \in \mathcal{J}} k(\boldsymbol{x}, X_j)} \left( \frac{A_i - \hat{\pi}(X_i)}{\hat{\pi}(X_i)(1 - \hat{\pi}(X_i))} \right. \tag{8}$$

$$\cdot \left\{ \mathbb{1}\{Y_i \leq y_{A_i}\} - \hat{F}_{Y|X,A_i}(y_{A_i}|X_i) \right\}$$

$$\left. + \hat{F}_a(y_1|X_i) - \hat{F}_0(y_0|X = X_i) \right)$$

where for any $\boldsymbol{x}' \in \mathcal{X}$, $a \in \{0,1\}$

$$\hat{\pi}(\boldsymbol{x}') := \sum_{i \in \mathcal{I}} \frac{k(\boldsymbol{x}', X_i)}{\sum_{j \in \mathcal{I}} k(\boldsymbol{x}', X_j)} \mathbb{1}\{A_i = 1\} \tag{9}$$

$$\hat{F}_a(y_a|\boldsymbol{x}') := \sum_{i \in \mathcal{I}} \frac{k(\boldsymbol{x}', X_i)}{\sum_{j \in \mathcal{I}} k(\boldsymbol{x}', X_j)\mathbb{1}\{A_j = a\}} \mathbb{1}\{Y_i \leq y_a\}\mathbb{1}\{A_i = a\}. \tag{10}$$

Note that for our theoretical results we do not specify how our nuisance parameters are estimated and the results only depend on the accuracy of our estimation of the nuisance parameters.

**Remark 6.** *We can re-write (8) as:*

$$\hat{h}(y_0, y_1|\boldsymbol{x}) := \sum_{i \in \mathcal{I}_1} \frac{k(\boldsymbol{x}, X_i)}{\sum_{i=1}^n k(\boldsymbol{x}, X_i)} \left( \frac{1}{\hat{\pi}(X_i)} \left\{ \mathbb{1}\{Y_i \leq y_1\} - \hat{F}_1(y_1|X_i) \right\} \right.$$

$$\left. + \hat{F}_1(y_1|X_i) - \hat{F}_0(y_0|X_i) \right)$$

$$- \sum_{i \in \mathcal{I}_0} \frac{k(\boldsymbol{x}, X_i)}{\sum_{i=1}^n k(\boldsymbol{x}, X_i)} \left( \frac{1}{1 - \hat{\pi}(X_i)} \left\{ \mathbb{1}\{Y_i \leq y_0\} - \hat{F}_0(y_0|X_i) \right\} \right.$$

$$\left. + \hat{F}_0(y_0|X_i) - \hat{F}_1(y_1|X_i) \right)$$

*Which can be helpful in terms of the practical implementation.*

### A.3 IPW pseudo-outcome estimator

Another pseudo-outcome one can use for estimating the treatment effect is the based on inverse propensity weighting. Specifically we can take our pseudo-outcome to be

$$\psi_{y_0,y_1}(Y', X', A') := \frac{A' - \pi(X')}{\pi(X')(1 - \pi(X'))} \mathbb{1}\{Y' \leq y_{A'}\}. \tag{11}$$

If we regress against it using NW estimation, and also use NW estimation for our nuisance parameter estimation as in (9) & (10), our estimate $\hat{h}$ will be increasing in $y_1$ and decreasing in $y_0$ meaning we do not need to perform any isotonic projection.

### A.4 Additional experimental details

As the method of Kallus and Oprescu [14] is one for estimating the CQTE, we transform it into an estimator of the CQC (which we denote by $\hat{g}$) using the following formula where $\hat{\tau}_q(.|.)$ is our CQTE estimator

$$\hat{g}(y|\boldsymbol{x}) = \hat{\tau}_q\left(F_{Y|X,0}(y|\boldsymbol{x})|\boldsymbol{x}\right) + y.$$

using the true CCDF, $F_{Y|X,0}$.

This is the inverse of equation (5) which defines the CQTE in terms of the CQC. Conversely we also tested transforming all our CQC estimators into CQTE estimators (using exactly equation (5) with $\hat{g}$ replacing $g^*$) and testing the accuracy on this space and found similar results.

Each experiment took no longer than 1 hour to run on a single 4 core CPU with 8GB of RAM.

The bandwidths of the kernels for the NW estimation of the nuisance parameters were chosen by a limited grid search on additional simulated data by validating against the true value of the nuisance parameters. While this is unrealistic in practice it was done to make estimation of the nuisance parameters as strong as possible. This was to err on the side of caution as strong nuisance parameter estimation would naturally favour the baseline approaches.

Code to implement our approach alongside Jupyter notebooks running our numerical experiments can be found in the supplementary materials. The code to implement the kernels is adapted from `https://github.com/wittawatj/kernel-gof/` which is free to use under the MIT Licence.

## B Additional results

### B.1 Simulation results

We run additional simulated experiments in order to further test and explore our approach. Throughout, our overall experimental set-up is the same as in Section 4.1

### B.1.1 10-dimensional example

To test our problem in higher dimensions we ran experiments where $X$ was 10-dimensional with $X$ uniform on $[-1,1]^{10}$. That is each component $X_j$ was independent with $X_j \sim U(-1,1)$. We then took

$$Y|X = \boldsymbol{x}, A = a \sim N(\sin(\gamma\pi(\boldsymbol{\beta}^\top \boldsymbol{x}), 1)$$
$$\pi(\boldsymbol{x}) = 0.4\sin(\gamma\pi(\boldsymbol{\beta}^\top \boldsymbol{x})) + 0.5$$

where $\boldsymbol{\beta}$ was randomly sampled from $N(0, (0.2)^2)$ and $\gamma \in [0,3]$. This gave the maximum gradient multiplied by $0.5 \operatorname{diam}(\mathcal{X}) = \sqrt{d}$ close to 1 making the rate of change of the CDFs and propensities similar to our 1-dimensional examples.

In our first experiment we take $2n = 1000$ and $\gamma \in \{0, 0.5, 1, 1.5, 2, 2.5, 3\}$. The results of this are shown in Figure 5a In our second experiment we take $\gamma = 1$ and $2n \in \{200, 500, 1000, 2000, 5000\}$. The results of this are shown in Figure 5b.

As we can see the DR approach performs the best with it performing close to oracle for low sample size and low frequency. As the frequency increases the DR estimator deviates from the oracle.

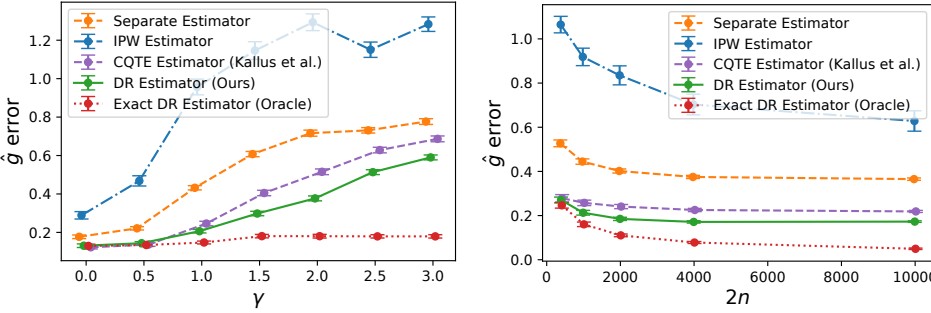

(a) Average error as $\gamma$ increases with 95% CIs for various approaches.

(b) Average error as sample size $n$ increases with 95% CIs for various approaches.

Figure 5: Estimation accuracy of 10-dimensional CQC estimation procedures for highly varying CCDFs and constant CQC with respect to $x$.

### B.1.2 Varying CQC

In this experiment our set-up is

$$Y|X, A = 0 \sim N\left(\frac{\sin(\gamma\pi x)}{0.5x + 1.5}, 1\right)$$
$$Y|X, A = 1 \sim N\left(\sin(\gamma\pi x) + 0.25x + 0.75, (0.5x + 1.5)^2\right)$$
$$\pi(x) = 0.4\sin(\gamma\pi x) + 0.5$$

for $\gamma \in [0, 10]$. This gives $g^* = (y + 0.5)(0.5x + 1.5)$ which is still simpler than the individual CCDFs but now does depend upon $x$.

In our first experiment we take $2n = 1000$ and $\gamma \in \{0, 2, 4, 6, 8, 10\}$. The results of this are shown in figure 5a In our second experiment we take $\gamma = 6$ and $2n \in \{200, 500, 1000, 2000, 5000\}$. The results of this are shown in Figure 5b.

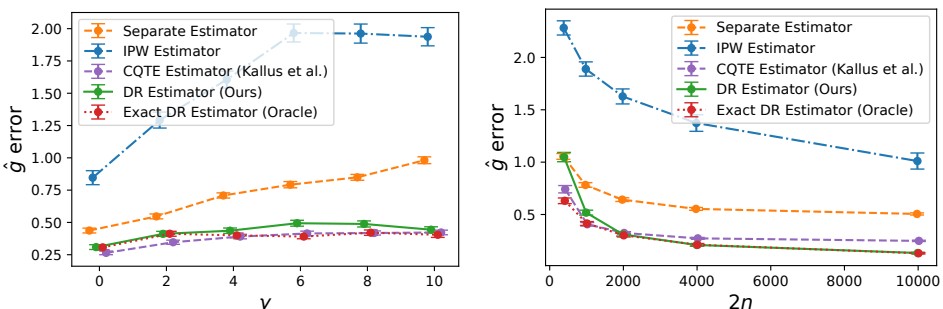

(a) Average error as $n$ increases with 95% CIs for various approaches.

(b) Average error as $n$ increases with 95% CIs for various approaches.

Figure 6: Estimation accuracy of CQC estimation procedures for highly varying CCDFs and linear CQC with respect to $x$.

### B.1.3 Constant $h^*$

In our previous examples, while $g^*$ has been simple and or constant, $h^*$ has actually included the high frequency sine term. In this experiment we adjust our original illustrative example to give a constant $h^*$ as well. Specifically we take

$$Y|X = x, A = 0 \sim \text{Unif}(\sin(\gamma\pi x), \sin(\gamma\pi x) + 1),$$
$$Y|X = x, A = 1 \sim \text{Unif}(2\sin(\gamma\pi x), 2\sin(\gamma\pi x) + 2).$$

for $\gamma \in [0, 10]$. We also take the propensity to be $\pi(x) = 0.4\sin(\gamma\pi x) + 0.5$. In this case $h^*(y_0, y_1|\boldsymbol{x}) = \frac{1}{2}y_1 - y_0$ which does not depend on $\boldsymbol{x}$ for any $y_0, y_1 \in \mathcal{Y}$.

In our first experiment we take $2n = 1000$ and $\gamma \in \{0, 2, 4, 6, 8, 10\}$. The results of this are shown in figure 5a In our second experiment we take $\gamma = 6$ and $2n \in \{200, 500, 1000, 2000, 5000\}$. The results of this are shown in Figure 5b. As we can see we obtain very similar results to original

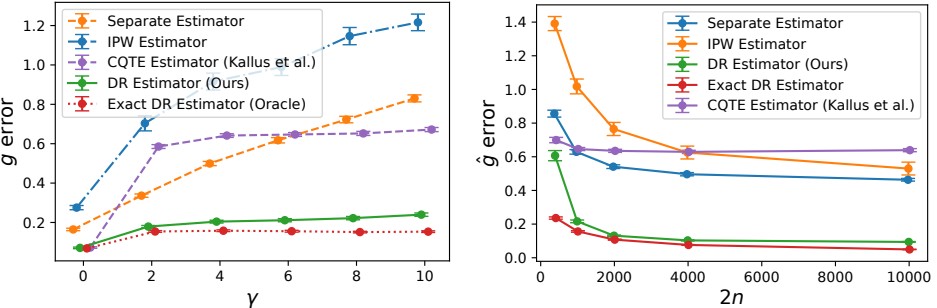

(a) Average error as $n$ increases with 95% CIs for various approaches.

(b) Average error as $n$ increases with 95% CIs for various approaches.

Figure 7: Constant $h^*$

Figure 8: Estimation accuracy of CQC estimation procedures for highly varying CCDFs and constant $h^*$ and CQC.

illustrative example suggesting that our method is effectively exploiting simplicity in $g^*$ rather than in $h^*$.

## B.2 Employment example: Comparing to the CQTE

To further explore the interpretability of our estimator we provide an estimate of the CQTE for comparison. This can be seen in Figure 9.

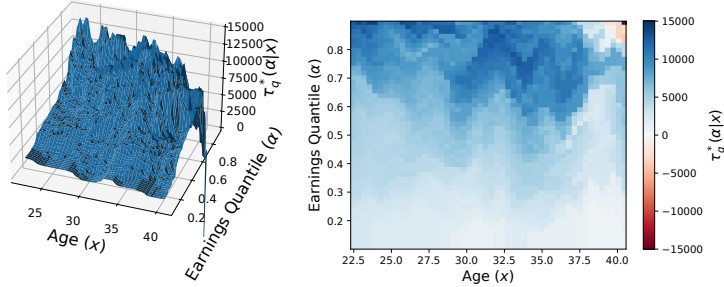

Figure 9: Surface and heat plot of the CQTE, $\tau_q^*(\alpha|\boldsymbol{x})$, for our employment data with $X =$Age, $\alpha=$Income Quantile.

In these plots we can see that the interpretation of the CQTE is less immediate than for the case of the CQC. This is because, for each covariate value, the quantile value corresponds to a different untreated response making comparisons between various values of the covariates less direct. From this plot we do still see that higher income quantiles are associated with a greater increase in wages. Interestingly the value of the CQTE plummets 90% at the highest quantile for individuals of age 40. This is likely an estimation error due to the limited data at higher quantiles as and higher ages.

## B.3 Colon cancer treatment

To demonstrate the effectiveness of our approach in medical settings, we apply it to a trial on the effect of colon cancer treatment on survival time/time to remission. This dataset was originally introduced in Laurie et al. [19] and can be found in the "survival" package in R and loaded with the line `data(colon, package="survival")`. In this dataset 929 are randomised to either receive Placebo or Levamisole. They are then followed up for a period of up to 3329 days and the time till

either death or recurrence of their cancer. For our analysis we take our response $(Y)$ as the time to first of death or recurrence. For simplicity, when an individual makes it to the end of a trial without an event and is censored we take that to be the time of their event. Looking at the data censoring times are mostly at around 2,000-3,000 days while over 90% of death or recurrence events that do occur, occur within 1,500 days. Therefore censored individuals will still have better responses than those who had events, as we would want. As our covariate, we took the patients' ages when joining the trial. We show a 3D plot and heat plot of the estimate of $\Delta(y|\boldsymbol{x}) = g^*(y|\boldsymbol{x}) - y$ in Figure 10.

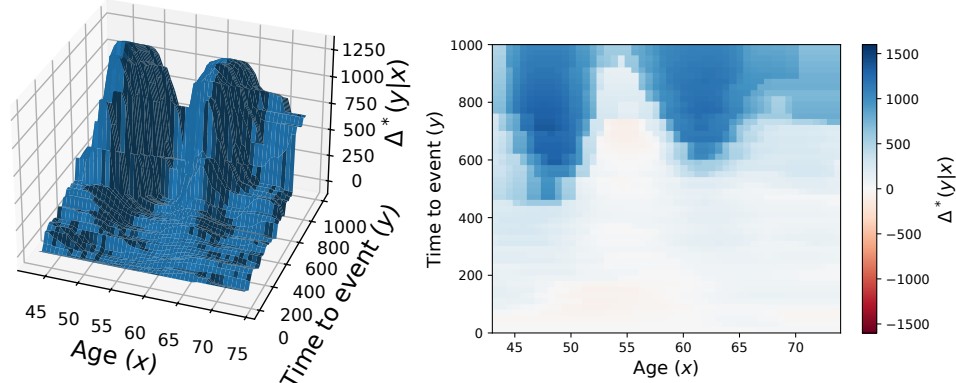

Figure 10: Surface plot and heat plot of $\Delta(y|\boldsymbol{x})$ over $y, \boldsymbol{x}$ for colon cancer trial data with $X =$Age, $Y =$Time to Event.

Interestingly from Figure 10 we see that the treatment appears to have little effect on patients aged 53-57. For the reaming patients, we can see that there is relatively little effect on the time to event for events of 400 days or less while at around 500 days the difference in time to event jumps to 1000 days. This jump takes the time to event (TTE) to the time when censoring begins. This gives evidence that rather than the treatment delaying the time to event, it increases the proportion of the patients who have no event during the trial essentially fully treating those patients for the duration of the trial. This shows the value of the CQC it was able to reveal meaningful information about the treatment beyond simply its positive effect. On top of this, we are able to identify the censoring within our results without explicitly controlling for it in any way.

## C  Additional theory

### C.1  Proof of $\hat{h}$ accuracy

We first define some additional notation. For an output $f$ which depends upon $z = (y, x, a)$ define $m_f(\boldsymbol{x}) = \mathbb{E}[f(Z)|X = \boldsymbol{x}]$ and

$$\hat{m}_f(\boldsymbol{x}) = \sum_{j \in \mathcal{J}} w_j f(Z_j),$$

with $w_j$ defined as before so that $\hat{m}_{\hat{\varphi}}(\boldsymbol{x})$ is our estimator and $\hat{m}_{\varphi}(\boldsymbol{x})$ is the oracle estimator.

Additionally define

$$\hat{b}(\boldsymbol{x}) \coloneqq m_{\hat{\varphi}-\varphi}(\boldsymbol{x}),$$

in other words the bias in our estimate of the pseudo-outcome for a given $\boldsymbol{x}$. We will also work with $\hat{m}_b(\boldsymbol{x})$ where we view $b$ as being a function of $z$.

Finally for $n \in \mathbb{N}$, $\delta \in (0, 1)$, define

$$\varepsilon_\delta(n) \coloneqq \begin{cases} \log(2/\delta)/n & \text{if } \delta \in (0, 2/e] \\ 1/n & \text{otherwise.} \end{cases}$$

**Theorem 3** (Stability Result). *For $\delta \in (0, 1)$ we have that with probability at least $1 - \delta$*

$$|\hat{m}_{\varphi}(\boldsymbol{x}) - m_{\varphi}(\boldsymbol{x})| \leq \mathbb{E}[\hat{m}_{\varphi}(\boldsymbol{x}) - m_{\varphi}(\boldsymbol{x})|X_{\mathcal{J}}, D_{\mathcal{I}}] \tag{12}$$

$$+ \sqrt{2\varepsilon_\delta(n)}\|\boldsymbol{w}\| \|\varphi - h(y_0, y_1|.)\|_{\boldsymbol{w}^2} + \frac{2\|\boldsymbol{w}\|_\infty \varepsilon_\delta(n)}{3\xi} =: \mathbb{B}(\varphi).$$

*Moreover, with probability at least $1 - \delta$ we have*

$$|\hat{m}_{\hat{\varphi}}(\boldsymbol{x}) - m_{\varphi}(\boldsymbol{x})| \leq \mathbb{B}(\varphi) + |\hat{m}_{\hat{b}}(\boldsymbol{x})| + \sqrt{2\varepsilon_\delta(n)}\|\boldsymbol{w}\|_2\|\hat{\varphi} - \varphi\|_{\boldsymbol{w}^2} =: \mathbb{B}^+(\hat{\varphi}). \qquad (13)$$

*Proof.* To prove (12) we first observe that since $\boldsymbol{w}$ is $D_{\mathcal{I}}, X_{\mathcal{J}}$-measurable we have

$$\mathbb{E}[\hat{m}_{\varphi}(\boldsymbol{x})^2|D_{\mathcal{I}}, X_{\mathcal{J}}] - \mathbb{E}[\hat{m}_{\varphi}(\boldsymbol{x})|D_{\mathcal{I}}, X_{\mathcal{J}}]^2 = \sum_{j \in \mathcal{J}} w_j^2 \mathbb{E}[(\varphi(Z_j) - \mathbb{E}[\varphi(Z)|X = X_j])^2|D_{\mathcal{I}}, X_{\mathcal{J}}]$$

Note also that we have $\max_{j \in \mathcal{J}} |w_j\{\varphi(Z_j) - \mathbb{E}[\varphi(Z_j)|D_{\mathcal{I}}, X_{\mathcal{J}}]\}| \leq \|\boldsymbol{w}\|_\infty 2\xi^{-1}$ almost surely. Note also that $\{\varphi(Z_j)\}_j \in \mathcal{J}$ are conditionally independent given $D_{\mathcal{I}}, X_{\mathcal{J}}$. Hence, the bound (12) follows from two applications Bernstein's inequality Boucheron et al. [7, Theorem 2.10], applied conditionally on $D_{\mathcal{I}}, X_{\mathcal{J}}$.

Next we note that by the triangle inequality we have

$$
\begin{aligned}
&|\mathbb{E}\{\hat{m}_{\hat{\varphi}}(\boldsymbol{x}) - m_{\varphi}(\boldsymbol{x}), | D_{\mathcal{I}}, X_{\mathcal{J}}\}| \\
&\leq |\mathbb{E}\{\hat{m}_{\hat{\varphi}}(\boldsymbol{x}) - \hat{m}_{\varphi}(\boldsymbol{x}) \,|\, D_{\mathcal{I}}, X_{\mathcal{J}}\}| + |\mathbb{E}\{\hat{m}_{\varphi}(\boldsymbol{x}) - m_{\varphi}(\boldsymbol{x}) \,|\, D_{\mathcal{I}}, X_{\mathcal{J}}\}| \\
&\leq \left| \sum_{j \in \mathcal{J}} w_j \mathbb{E}\left[\hat{\varphi}(Z_j) - \varphi(Z_j)|D_{\mathcal{I}}, X_{\mathcal{J}}\right] \right| + |\mathbb{E}\{\hat{m}_{\varphi}(\boldsymbol{x}) - m_{\varphi}(\boldsymbol{x}) \,|\, D_{\mathcal{I}}, X_{\mathcal{J}}\}| \\
&= \|\hat{m}_{\hat{b}}\|_{\boldsymbol{w},1} + |\mathbb{E}\{\hat{m}_{\varphi}(\boldsymbol{x}) - m_{\varphi}(\boldsymbol{x}) \,|\, D_{\mathcal{I}}, X_{\mathcal{J}}\}|
\end{aligned}
$$

Moreover, since $\mathbb{E}[\hat{\varphi}(Z_j)|D_{\mathcal{I}}, X_{\mathcal{J}}]$ is the projection of $\hat{\varphi}(Z_j)$ onto the subspace of $D_{\mathcal{I}}, X_{\mathcal{J}}$-measureable functions we have

$$
\begin{aligned}
&\|\boldsymbol{w}\|_2^2 \|\hat{m}_{\hat{\varphi}} - m_{\hat{\varphi}}(\boldsymbol{x})\|_{\boldsymbol{w}^2}^2 \\
&= \sum_{j \in \mathcal{J}} w_j^2 \, \mathbb{E}\{(\hat{\varphi}(Z) - \mathbb{E}\{\hat{\varphi}(Z)|X = X_j\})^2|D_{\mathcal{I}}, X_{\mathcal{J}}\} \\
&\leq \sum_{i \in [n]} w_j^2 \, \mathbb{E}\{(\hat{\varphi}(Z_j) - \mathbb{E}\{\varphi(Z_j)|D_{\mathcal{I}}, X_{\mathcal{J}}\})^2|D_{\mathcal{I}}, X_{\mathcal{J}}\} \\
&= \sum_{i \in [n]} w_j^2 \, \mathbb{E}\{(\{(\hat{\varphi}(Z_j) - \varphi(Z_j)) + (\varphi(Z_j) - \mathbb{E}\{\varphi(Z_j)|D_{\mathcal{I}}, X_{\mathcal{J}}\})\}^2|D_{\mathcal{I}}, X_{\mathcal{J}}) \\
&= \sum_{i \in [n]} w_j^2 \, (\mathbb{E}\{(\hat{\varphi}(Z_j) - \varphi(Z_j))^2|D_{\mathcal{I}}, X_{\mathcal{J}}\} + \mathbb{E}\{(\varphi(Z_j) - \mathbb{E}\{\varphi(Z_j)|D_{\mathcal{I}}, X_{\mathcal{J}}\})^2|D_{\mathcal{I}}, X_{\mathcal{J}}\}) \\
&= \|\boldsymbol{w}\|_2^2 \{(\|\hat{m}_{\hat{\varphi}} - \hat{m}_{\varphi}\|_{\boldsymbol{w}^2}^2 + \|\hat{m}_{\varphi} - m_{\varphi}\|_{\boldsymbol{w}^2}^2\} \\
&\leq \|\boldsymbol{w}\|_2^2 \{(\|\hat{m}_{\hat{\varphi}} - \hat{m}_{\varphi}\|_{\boldsymbol{w}^2} + \|\hat{m}_{\varphi} - m_{\varphi}\|_{\boldsymbol{w}^2}\}^2.
\end{aligned}
$$

Hence, to deduce (13) we apply the first bound with $\hat{\varphi}$ in place of $\varphi$ to obtain the following bound with probability at least $1 - \delta$,

$$
\begin{aligned}
|\hat{m}_{\varphi}(\boldsymbol{x}) - m_{\varphi}(\boldsymbol{x})| \leq & |\mathbb{E}\{\hat{m}_{\hat{\varphi}}(\boldsymbol{x}) - m_{\varphi}(\boldsymbol{x}) \,|\, D_{\mathcal{I}}, X_{\mathcal{J}}\}| + \sqrt{2\varepsilon_\delta(n)} \|\boldsymbol{w}\|_2 \|\hat{m}_{\hat{\varphi}} - m_{\hat{\varphi}}\|_{\boldsymbol{w}^2} \\
& + 2\xi^{-1}\|\boldsymbol{w}\|_\infty \varepsilon_\delta(n)/3 \\
\leq & \|\hat{m}_{\hat{b}}\|_{\boldsymbol{w},1} + |\mathbb{E}\{\hat{m}_{\varphi}(\boldsymbol{x}) - m_{\varphi}(\boldsymbol{x}) \,|\, D_{\mathcal{I}}, X_{\mathcal{J}}\}| \\
& + \sqrt{2\varepsilon_n(\delta)}\|\boldsymbol{w}\|_2 \{(\|\hat{m}_{\hat{\varphi}} - \hat{m}_{\varphi}\|_{\boldsymbol{w}^2} + \|\hat{m}_{\varphi} - m_{\varphi}\|_{\boldsymbol{w}^2}\} \\
& + 2\xi^{-1}\|\boldsymbol{w}\|_\infty \varepsilon_\delta(n)/3 \\
= & \mathbb{B}(\varphi) + \|\hat{m}_{\hat{b}}\|_{\boldsymbol{w},1} + \sqrt{2\varepsilon_\delta(n)} \|\boldsymbol{w}\|_2 \|\hat{m}_{\hat{\varphi}} - \hat{m}_{\varphi}\|_{\boldsymbol{w}^2} \\
= & \mathbb{B}^+(\hat{\varphi}),
\end{aligned}
$$

as required. $\qquad\square$

We apply the following result from Kennedy [15].

**Proposition 4** (Proposition 2 from Kennedy [15])**.** *Suppose that $\hat{b}(\boldsymbol{x}) = \hat{b}_1(\boldsymbol{x})\hat{b}_2(\boldsymbol{x})$ then*

$$|\hat{m}_{\hat{b}}(\boldsymbol{x})| = \|\boldsymbol{w}\|_1 \|\hat{b}_1\|_{\boldsymbol{w}} \|\hat{b}_1\|_{\boldsymbol{w}}.$$

*Proof.* Follows from the Cauchy-Schwartz inequality. □

**Proposition 5.** *Our pseudo-outcome is conditionally unbiased,*

$$\mathbb{E}[\varphi_{y_0,y_1}(Z)|X = \boldsymbol{x}] = h^*(y_0, y_1|\boldsymbol{x}).$$

*Proof.* We have

$$\mathbb{E}[\varphi_{y_0,y_1}(Z)|X = \boldsymbol{x}, A = 1] = \frac{1}{\pi(\boldsymbol{x})}\Big(\mathbb{E}[\mathbb{1}\{Y \le y_1\}|X = \boldsymbol{x}, A = 1] - F_1(y_1|\boldsymbol{x})\Big) + h^*(y_0, y_1|\boldsymbol{x})$$
$$= h^*(y_0, y_1|\boldsymbol{x}),$$

and similarly $\mathbb{E}[\varphi_{y_0,y_1}(Z)|X = \boldsymbol{x}, A = 1] = h^*(y_0, y_1|\boldsymbol{x})$. The result now follows by the law of total expectation. □

### C.1.1 Proof of Proposition 1

*Proof of Proposition 1.* Fix $y_0, y_1, \boldsymbol{x}$ Let $\hat{m}_{\hat{\varphi}}(\boldsymbol{x})$ be the regression of $X$ against the estimated pseudo-outcome $\hat{\varphi}_{y_0,y_1}(Z)$ evaluated at $\boldsymbol{x}$ (i.e. $\hat{h}(y_0, y_1|\boldsymbol{x})$), let $\hat{m}_{\varphi}$ be the same but with the estimated pseudo-outcome replaced with the exact pseudo-outcome $\varphi_{y_0,y_1}$. Finally take

$$m_{\varphi}(\boldsymbol{x}) := \mathbb{E}[\varphi_{y_0,y_1}(Z)|X = \boldsymbol{x}] = \mathbb{P}(Y \le y_1|X = \boldsymbol{x}, A = 1) - \mathbb{P}(Y \le y_1|X = \boldsymbol{x}, A = 0).$$

Theorem 3 gives us that with probability. at least $1 - \delta$

$$|\hat{m}_{\hat{\varphi}}(\boldsymbol{x}) - m_{\varphi}(\boldsymbol{x})| \le \varepsilon_{\gamma}(n, \delta) + |\hat{m}_{\hat{b}}(\boldsymbol{x})| + \sqrt{2\varepsilon_{\delta}(n)}\|\boldsymbol{w}\|_2 \|\hat{\varphi} - \varphi\|_{\boldsymbol{w}^2}$$
$$\le \varepsilon_{\gamma}(n, \delta) + |\hat{m}_{\hat{b}}(\boldsymbol{x})| + \sqrt{2\varepsilon_{\delta}(n)}\|\hat{\varphi} - \varphi\|_{\boldsymbol{w}^2}$$

with $\hat{b}((y, \boldsymbol{x}, a)) := \mathbb{E}[\hat{\varphi}(Z) - \varphi(Z)|X = \boldsymbol{x}]$.

To bound $\hat{b}$ we have that from Proposition 5,

$$\mathbb{E}[\varphi_{y_0,y_1}(Z)|X = \boldsymbol{x}] = \mathbb{P}(Y \le y_1|X = \boldsymbol{x}, A = 1) - \mathbb{P}(Y \le y_0|X = \boldsymbol{x}, A = 0).$$

Additionally, by splitting over the events $\{A = 1\}, \{A = 0\}$ and noting that $\mathbb{P}(A = 1|X = \boldsymbol{x}) = \pi(\boldsymbol{x})$ we have

$$\mathbb{E}[\hat{\varphi}_{y_0,y_1}(Z)|X = \boldsymbol{x}] = \left(\frac{\pi(\boldsymbol{x})}{\hat{\pi}(\boldsymbol{x})}\right)\left(\mathbb{P}(Y \le y_1|X = \boldsymbol{x}, A = 1) - \hat{\mathbb{P}}(Y \le y_1|X = \boldsymbol{x}, A = 1)\right)$$
$$- \frac{1 - \pi(\boldsymbol{x})}{1 - \hat{\pi}(\boldsymbol{x})}\left(\mathbb{P}(Y \le y_0|X = \boldsymbol{x}, A = 0) - \hat{\mathbb{P}}(Y \le y_0|X = \boldsymbol{x}, A = 0)\right)$$
$$+ \hat{\mathbb{P}}(Y \le y_1|X = \boldsymbol{x}, A = 1) - \hat{\mathbb{P}}(Y \le y_0|X = \boldsymbol{x}, A = 0).$$

Hence

$$\hat{b}(\boldsymbol{x}) = \left(\frac{\pi(\boldsymbol{x})}{\hat{\pi}(\boldsymbol{x})} - 1\right)\left(\mathbb{P}(Y \le y_1|X = \boldsymbol{x}, A = 1) - \hat{\mathbb{P}}(Y \le y_1|X = \boldsymbol{x}, A = 1)\right)$$
$$- \left(\frac{1 - \pi(\boldsymbol{x})}{1 - \hat{\pi}(\boldsymbol{x})} - 1\right)\left(\mathbb{P}(Y \le y_0|X = \boldsymbol{x}, A = 0) - \hat{\mathbb{P}}(Y \le y_0|X = \boldsymbol{x}, A = 0)\right).$$

We have that using Proposition 4

$$|\hat{m}_{\hat{b}}(\boldsymbol{x})| \leq \|\boldsymbol{w}\|_1 \left\| \frac{\pi}{\hat{\pi}} - 1 \right\|_{\boldsymbol{w}} \left\| \mathbb{P}(Y \leq y_1|X, A = 1) - \hat{\mathbb{P}}(Y \leq y_1|X, A = 1) \right\|_{\boldsymbol{w}}$$

$$+ \left\| \frac{1-\pi}{1-\hat{\pi}} - 1 \right\|_{\boldsymbol{w}} \left\| \mathbb{P}(Y \leq y_0|X, A = 0) - \hat{\mathbb{P}}(Y \leq y_0|X, A = 0) \right\|_{\boldsymbol{w}}$$

$$\leq \frac{1}{\xi} \sum_{a=0}^{1} \|\pi - \hat{\pi}\|_{\boldsymbol{w}} \left\| \mathbb{P}(Y \leq y_a|X, A = a) - \hat{\mathbb{P}}(Y \leq y_a|X, A = a) \right\|_{\boldsymbol{w}}.$$

Now it is simply a matter of bounding all the relevant terms which we do through the following events

$$E_{\text{oracle}} := \left\{ |\hat{m}_{\hat{\varphi}} - m_\varphi| \leq |\hat{m}_\varphi(\boldsymbol{x}) - m_\varphi(\boldsymbol{x})| + |\hat{m}_{\hat{b}}(\boldsymbol{x})| + \sqrt{2\varepsilon_\delta(n)} \|\boldsymbol{w}\|_2 \|\hat{\varphi} - \varphi\|_{\boldsymbol{w}^2} \right\}$$

$$E_{\hat{\varphi}} := \left\{ \|\hat{\varphi} - \varphi\|_{\boldsymbol{w}^2} \leq \varepsilon_{\hat{\varphi}}(n, \delta/4) \right\}$$

$$E_\gamma := \left\{ \hat{m}_\varphi - m_\varphi \leq \varepsilon_\gamma(n, \delta/4) \right\}$$

$$E_\alpha := \left\{ \|\pi - \hat{\pi}\|_{\boldsymbol{w},2} \leq \varepsilon_\alpha(n, \delta/4) \right\}$$

$$E_{\beta,0} := \left\{ \|\hat{\mathbb{P}}(Y \leq y_0|X, A = 0) - \mathbb{P}(Y \leq y_0|X, A = 0)\|_{\boldsymbol{w},2} \leq g_\beta(n, \delta/4) \right\}$$

$$E_{\beta,1} := \left\{ \|\hat{\mathbb{P}}(Y \leq y_1|X, A = 1) - \mathbb{P}(Y \leq y_1|X, A = 1)\|_{\boldsymbol{w},2} \leq \varepsilon_\beta(n, \delta/4) \right\}$$

According to our assumptions and previous results, $E_{\text{oracle}}, E_{\hat{\varphi}}, E_\gamma, E_\alpha \cap E_{\beta,0} \cap E_{\beta,1}$ separately occur w.p. at least $1 - \delta/4$. Then by the union bound, the intersection of all these events holds w.p. at least $1 - \delta$ and under these events

$$|\hat{m}_{\hat{\varphi}} - \hat{m}_\varphi + \hat{m}_\varphi - m_\varphi| \leq \varepsilon_{T_n}(n, \delta/4) + \frac{2}{\xi}\varepsilon_\alpha(n, \delta/4)\varepsilon_\beta(n, \delta/4) + \varepsilon_\gamma(n, \delta/4)\varepsilon_{\boldsymbol{w}}(n, \delta/4)$$

Where the first term comes from Proposition 3, the second from Proposition 4 and the final term from smoothness of $g^*$. $\square$

## C.2 Proof of $\hat{g}$ accuracy

**Proposition 6** (Maximum step-size bound). *Suppose that both the outer regression and estimation of CCDFs is fit using linear smoothers so that*

$$\hat{h}(y_0, y_1|\boldsymbol{x}) = \sum_{j \in \mathcal{J}} w_j(\boldsymbol{x}; X_{\mathcal{J}})\hat{\varphi}(Z_j) \qquad \hat{F}_a(y|\boldsymbol{x}') = \sum_{i \in \mathcal{I}} w_{F_a;i}(\boldsymbol{x}'; X_{\mathcal{J}})\mathbb{1}\{Y_i \leq y\}$$

*with $w_j(\boldsymbol{x}), w_i(\boldsymbol{x}')$ Additionally suppose with probability at least $1 - \delta$*

$$\max \left\{ \max_{j \in \mathcal{J}} w_j(\boldsymbol{x}), \max_{i \in \mathcal{I}} w_{F_a;i}(X) \right\} \leq \varepsilon_{\boldsymbol{w}}(n, \delta).$$

*Then with probability at least $1 - \delta$*

$$\max_{j \in [n_1+1]_0} |\hat{h}(y_0, Y_1^{(j)}|\boldsymbol{x}) - \hat{h}(y_0, Y_1^{(j+1)}|\boldsymbol{x})| \geq \xi^{-1}\varepsilon_{\boldsymbol{w}}(n, \delta/n)$$

*where $[n_1 + 1]_0 := [n_1 + 1] \cup \{0\}$, $\hat{h}(y_0, Y_1^{(0)}|\boldsymbol{x}) = -1$, $\hat{h}(y_0, Y_1^{(n_1+1)}|\boldsymbol{x})$, and $\{Y^{(i)}\}_{i \in I_1}$ is the sorted version of $\{Y_i\}_{i \in I_1}$.*

*Proof.* We immediately have via union bounds that with probability at least $1 - \delta$

$$\max \left\{ \max_{j \in \mathcal{J}} w_j(\boldsymbol{x}), \max_{j \in \mathcal{J}} w_{F_a;i}(X_j) \right\} \leq \varepsilon_{\boldsymbol{w}}(n, \delta/n).$$

Now we can try to bound the jump size of our function. To do so for a step function $f : \mathcal{Y} \to \mathbb{R}$ and a step points $y \in \mathcal{Y}$ we define

$$f(\Delta y) := \lim_{\varepsilon \downarrow 0} f(y) - f(y - \varepsilon),$$

i.e. the size of the step at $y$. From the definition of our pseudo-outcome, the jumps occur in different forms at $\{Y_j\}_{\{j\in\mathcal{J}, A_j=1\}}$, $\{Y(i)\}_{\{i\in\mathcal{I}|A^{(i)}=1\}}$. Note that we are assuming $Y$ continuous so these points are all distinct a.s. .

For $Y_j$ with $j\in\mathcal{J}$ and $A_j=1$. We have

$$\hat{h}(y_0,\Delta Y_j|\boldsymbol{x}) = w_{\varphi,j}\frac{1}{\hat{\pi}(X_j)}\mathbb{1}\{Y_j\le\Delta Y_j\}$$
$$\le\frac{w_{\varphi,j}}{\xi}.$$

For $i\in\mathcal{I}$ with $A^{(i)}=1$, we have that

$$|\hat{h}(y_0,\Delta Y_i|\boldsymbol{x})| = \left|\sum_{j\in\mathcal{J}}w_j\left(-\mathbb{1}\{A_j=1\}\frac{1}{\hat{\pi}(X_j)}\hat{F}_1(\Delta Y_i|X_j)+\hat{F}_1(\Delta Y_i|X_j)\right)\right|$$
$$\le\frac{1}{\xi}\max_{j\in\mathcal{J}}\hat{F}_1(\Delta Y_i|X_j)$$
$$\le\frac{1}{\xi}\max_{j\in\mathcal{J}}w_{F_a;i}(X_j)\mathbb{1}\{Y_i\le\Delta Y_i\}$$
$$=\frac{1}{\xi}\max_{j\in\mathcal{J}}w_{F_a;i}(X_j).$$

Hence the maximum jump size is less than

$$\frac{1}{\xi}\max\left\{\max_{j\in\mathcal{J}}w_{\varphi,j}(\boldsymbol{x}),\max_{i\in\mathcal{I},j\in\mathcal{J}}w_{F_a;i}(X_j)\right\}.$$

Therefore combining this with our previous bounds gives that with probability at least $1-\delta$

$$\max_{i\in[n_1+1]_0}|\hat{h}(y_0,Y^{(i)}|\boldsymbol{x})-\hat{h}(y_0,Y^{(i+1)}|\boldsymbol{x})|\le\xi^{-1}\varepsilon_{\boldsymbol{w}}(n,\delta/n)$$

$\square$

$F_1$ We also have a result ensuring the step size of the projected function is even smaller.

**Proposition 7.** *Let $\boldsymbol{\alpha}\in\mathbb{R}^p$ and $\tilde{\boldsymbol{\alpha}}:=P_{\mathrm{Iso}(p)}(\boldsymbol{\alpha})$. Then $|\alpha_l-\alpha_{l+1}|\ge|\tilde{\alpha}_l-\tilde{\alpha}_{l+1}|$ for all $l\in[p]$.*

*Proof.* We will prove this by first giving an algorithm to compute the projection. Define

$$\psi_l(\boldsymbol{\alpha}):=\begin{cases}\boldsymbol{\alpha} & \text{if }\alpha_l\le\alpha_{l+1}\\ \left(\alpha_1,\dots,\frac{\alpha_l+\alpha_{l+1}}{2},\frac{\alpha_l+\alpha_{l+1}}{2},\alpha_m\right) & \text{if }\alpha_l>\alpha_l+1.\end{cases}$$

Now define $\boldsymbol{\alpha}^{(0,0)}=\boldsymbol{\alpha}$, $\boldsymbol{\alpha}^{(n,i)}=\psi_i(\alpha^{(n,i-1)})$ for $i\in[p-1]$ and $\boldsymbol{\alpha}^{(n,0)}=\boldsymbol{\alpha}^{(n-1,p-1)}$ for $n\in\mathbb{N}$.

Finally define $\boldsymbol{\alpha}^{(t)}=\boldsymbol{\alpha}^{(\lfloor t/p\rfloor,\mathrm{mod}(t,p)}$. Then by Yang and Barber [32] we know that $\lim_{t\to\infty}\boldsymbol{\alpha}^{(t)}=P_{\mathrm{Iso}(p)}(\boldsymbol{\alpha})$.

Now if $|\alpha_{l+1}^{(n,l+1)}-\alpha_l^{(n,l+1)}|>|\alpha_{l+1}^{(n,l)}-\alpha_l^{(n,l)}|$. Then $\alpha_{l+1}^{(n,l+1)}$ has be moved by $\psi_{l+1}$. Hence $\alpha_{l+1}^{(n,l+1)}<\alpha_{l+1}^{(n,l)}$. For this to increase the distance then we must have

$$\alpha_{l+1}^{(n,l+1)}<\alpha_l^{(n,l+1)}.$$

Therefore as $\alpha_l^{(n+1,l-1)}\ge\alpha_l^{n,l+1}$, we have that

$$\alpha_{l+1}^{(n+1,l)}=\alpha_l^{(n+1,l)}.$$

By a similar (simpler) argument $|\alpha_l^{(n,l-1)}-\alpha_{l+1}^{(n,l-1)}|>|\alpha_l^{(n,l-2)}-\alpha_{l+1}^{(n,l-2)}|$ then $\alpha_l^{(n,l)}=\alpha_{l+1}^{(n,l)}$. Hence if any iteration moves adjacent points further apart, a later iteration will make them equal meaning that on convergence they will be equal. Hence we have proved our claim. $\square$

We now have the result which justifies our choice to take the isotonic projection of the data which is take from Yang and Barber [32].

**Proposition 8** (Theorem 1 from Yang and Barber [32]). *Let $\boldsymbol{z}^*, \hat{\boldsymbol{z}} \in \mathbb{R}^p$ with $\boldsymbol{z}^* \in \mathrm{Iso}(p)$. Then for $\tilde{\boldsymbol{z}} := P_{\mathrm{Iso}(p)}(\hat{\boldsymbol{z}})$,*

$$\max_l |z_l^* - \tilde{z}_l| \leq \max_l |z_l^* - \hat{z}_l|.$$

We now use this isotonic projection to obtain supremum bounds on the accuracy of our estimator.

**Proposition 9** (Supremum bound on $\hat{h}$ accuracy). *Fix $\boldsymbol{x} \in \mathcal{X}, y \in y_0$ and let $\hat{h}$ be our original estimate of $h^*$. For $m \in \mathbb{N}$, take $\{Y^{(l)}\}_{l=1}^m$ to be a potentially random set of points in $\mathcal{Y}$ in increasing order.*

*Now define $\boldsymbol{\alpha} \in \mathbb{R}^m$ by $\alpha_l := \hat{h}(y_0, Y^{(l)}|\boldsymbol{x})$ and $\tilde{\boldsymbol{\alpha}} := P_{\mathrm{Iso}(m)}(\boldsymbol{\alpha})$. Finally, define $\tilde{h}$ to be the piecewise constant right continuous function with $\tilde{h}(y_0, Y^{(l)}|\boldsymbol{x}) := \tilde{\alpha}_l$.*

*Suppose for any $y_1 \in \mathcal{Y}$, $n \in \mathbb{N}$, and $\delta, \delta' > 0$ that*

$$\mathbb{P}\left(\left|\tilde{h}(y_0, y_1|\boldsymbol{x}) - h^*(y_0, y_1|\boldsymbol{x})\right| \geq \varepsilon_h(n, \delta)\right) \leq \delta$$

$$\mathbb{P}\left(\max_{l \in [m]_0} \left|\hat{h}(y_0, Y^{(l)}|\boldsymbol{x}) - \hat{h}(y_0, Y^{(l+1)}|\boldsymbol{x})\right| \geq \varepsilon_{step}(n, m, \delta)\right) \leq \delta$$

*where we take $[m]_0 := [m] \cup \{0\}$, $\hat{h}(y_0, Y^{(0)}|\boldsymbol{x}) := -1$, and $\hat{h}(y_0, Y^{(m+1)}|\boldsymbol{x}) := 1$.*

*Then for any $\delta, \delta' > 0, n \in \mathbb{N}$,*

$$\mathbb{P}\left(\sup_{y_1 \in \mathcal{Y}} \left|\hat{h}(y_0, y_1|\boldsymbol{x}) - h^*(y_0, y_1|\boldsymbol{x})\right| \leq \varepsilon_h(n, \delta/m) + \varepsilon_{step}(n, m, \delta')\right) \geq 1 - \delta - \delta'.$$

*Proof.* For each $l \in [m]$ define the event

$$E_{h,l} := \left\{\left|\hat{h}(y_0, Y_1^{(l)}|\boldsymbol{x}) - h^*(y_0, Y_1^{(l)}|\boldsymbol{x})\right| \leq \varepsilon_h(n, \delta/m)\right\}$$

and take $E_h := \bigcap_{l=1}^n E_{h,l}$. Additionally define the event

$$E_{step} := \left\{\max_{l \in [m]_0} \left|\hat{h}(y_0, Y^{(l)}|\boldsymbol{x}) - \hat{h}(y_0, Y^{(l+1)}|\boldsymbol{x})\right| < \varepsilon_{step}(n, m, \delta)\right\}$$

Then $E_h$ and $E_{step}$ hold w.p.s at least $1 - \delta$ and $1 - \delta'$ respectively. Also under $E_h, E_{step}$, by Propositions 7 & 8, we have that

$$\max_{l \in [m]} \left|\tilde{h}(y_0, Y_1^{(l)}|\boldsymbol{x}) - h^*(y_0, Y_1^{(l)}|\boldsymbol{x})\right| \leq \varepsilon_h(n, \delta/m)$$

$$\max_{l \in [m]_0} \left|\tilde{h}(y_0, Y^{(l)}|\boldsymbol{x}) - \tilde{h}(y_0, Y^{(l+1)}|\boldsymbol{x})\right| < \varepsilon_{step}(n, m, \delta)$$

We then have that for any $y_1 \in \mathcal{Y}$ there exists $i \in [m]$ such that $y_1^{(l-1)} \leq y_1 \leq y_1^{(l)}$.

Hence by monotonicity, we have the following 2 inequalities

$$\hat{h}(y_0, y_1|\boldsymbol{x}) - h^*(y_0, y_1|\boldsymbol{x}) \geq \underbrace{\tilde{h}(y_0, Y_1^{(l-1)}|\boldsymbol{x}) - h^*(y_0, Y_1^{(l)}|\boldsymbol{x})}_{\Lambda_1}$$

$$\hat{h}(y_0, y_1|\boldsymbol{x}) - h^*(y_0, y_1|\boldsymbol{x}) \leq \underbrace{\tilde{h}(y_0, Y_1^{(l)}|\boldsymbol{x}) - h^*(y_0, Y_1^{(l-1)}|\boldsymbol{x})}_{\Lambda_2}.$$

Now for the first inequality we have

$$\Lambda_1 = \tilde{h}(y_0, Y_1^{(l-1)}|\boldsymbol{x}) - \tilde{h}^*(y_0, Y_1^{(l)}|\boldsymbol{x}) + \tilde{h}(y_0, Y_1^{(l)}|\boldsymbol{x}) - h^*(y_0, Y_1^{(l)}|\boldsymbol{x})$$
$$\geq -\varepsilon_h(n, \delta) - \varepsilon_{step}(n, m, \delta').$$

With the final inequality coming from $E_{h,l-1}$ and our definition of $h$. By a similar argument we get from $E_{h,l}$ that $\Lambda_2 \leq \varepsilon_h(n,\delta) + \varepsilon_{step}(n,m,\delta')$. Again we can use the same approach to get that under $E_{h,l-1}, E_{h,l}, E_{step}$

$$-\varepsilon_h(n,\delta) - \varepsilon_{step}(n,m,\delta') \leq h^*(y_0, y_1|\boldsymbol{x}) - \tilde{h}(y_0, y_1|\boldsymbol{x}) \leq \varepsilon_h(n,\delta) + \varepsilon_{step}(n,m,\delta').$$

Hence for our specific $y_1$ under $E_{h,l}, E_{h,l-1}, E_{step}$

$$\left| h^*(y_0, y_1|\boldsymbol{x}) - \tilde{h}(y_0, y_1|\boldsymbol{x}) \right| \leq \varepsilon_h(n,\delta) + \varepsilon_{step}(n,m,\delta')$$

Now as this inequality holds for arbitrary $y_1$ under $E_h \cap E_{step}$ (an event which not depend on our choice of $y_1$) and the intersection of these two events holds w.p. at least $1 - \delta - \delta'$ by the union bound we have our result. $\qquad\square$

Our final key result before we can piece them all together will be to obtain accuracy in $g$ from our accuracy in $h^*$

**Theorem 10** (Single point accuracy bound). *Fix $\boldsymbol{x}, y_0$, assume that $h, \tilde{h}$ are strictly monotonic in $y_1$ and suppose that*

$$\sup_{y_1} |h^*(y_0, y_1|\boldsymbol{x}) - \tilde{h}(y_0, y_1|\boldsymbol{x})| < \varepsilon.$$

*Additionally let $\alpha := F_0(y_0|\boldsymbol{x})$ and assume that $f_{Y|X,A=1}(y_1|\boldsymbol{x}) > \eta$ for all $y_1 \in F_1^{-1}(B_\varepsilon(\alpha)|\boldsymbol{x})$ where here we are taking $F_1^{-1}(A|\boldsymbol{x})$ to be the pre-image in $\mathcal{Y}$ of the set $A$ for fixed $\boldsymbol{x}$.*

*Then*

$$|g^*(y_0|\boldsymbol{x}) - \hat{g}(y_0|\boldsymbol{x})| \leq \frac{2\varepsilon}{\eta},$$

*where $g^*(y_0|\boldsymbol{x})$ is the unique $y_1$ such that $h^*(y_0, y_1|\boldsymbol{x}) = 0$.*

*Proof.* Let $y_1^* := g^*(y_0|\boldsymbol{x})$ and $\hat{y}_1 := \hat{g}(y_0|\boldsymbol{x})$ so that $F_1(y_1^*|\boldsymbol{x})$. From our accuracy assumption on $\hat{h}$ in the set-up of the Theorem we have

$$\begin{aligned}
|F_1(y_1^*|\boldsymbol{x}) - F_1(\hat{y}_1|\boldsymbol{x})| &= |h^*(y_0, y_1^*|\boldsymbol{x}) - h^*(y_0, \hat{y}_1|\boldsymbol{x})| \\
&= |h^*(y_0, \hat{y}_1|\boldsymbol{x})| \\
&\leq |h^*(y_0, \hat{y}_1|\boldsymbol{x}) - \hat{h}(y_0, \hat{y}_1|\boldsymbol{x})| + |\hat{h}(y_0, \hat{y}_1|\boldsymbol{x})| \\
&\leq |h^*(y_0, \hat{y}_1|\boldsymbol{x}) - \hat{h}(y_0, \hat{y}_1|\boldsymbol{x})| + |\hat{h}(y_0, y_1^*|\boldsymbol{x})| \\
&\leq |h^*(y_0, \hat{y}_1|\boldsymbol{x}) - \hat{h}(y_0, \hat{y}_1|\boldsymbol{x})| + |\hat{h}(y_0, y_1^*|\boldsymbol{x}) - h^*(y_0, y_1^*|\boldsymbol{x})| \\
&\leq 2\varepsilon,
\end{aligned}$$

where the second and fifth line come from the definition of $y_1^*$ and the fourth from the definition of $\hat{y}_1$.

If we define $\partial_k f$ to be the derivative with respect to the $k^{\text{th}}$ argument of $f$ then we have

$$\begin{aligned}
|\hat{g}(y_0|\boldsymbol{x}) - g^*(y_0|\boldsymbol{x})| &= |F_1^{-1}(F_1(y_1^*|\boldsymbol{x})|\boldsymbol{x}) - F_1^{-1}(F_1(\hat{y}_1|\boldsymbol{x})|\boldsymbol{x})| \\
&= \left| \int_{F_1(y_1^*|\boldsymbol{x})}^{F_1(\hat{y}_1|\boldsymbol{x})} \partial_1 F_1^{-1}(\beta|\boldsymbol{x}) \mathrm{d}\beta \right| \\
&= \left| \int_{F_1(y_1^*|\boldsymbol{x})}^{F_1(\hat{y}_1|\boldsymbol{x})} \frac{1}{f((F_1^{-1}(\beta|\boldsymbol{x})|\boldsymbol{x})} \mathrm{d}\beta \right| \\
&\leq 2\varepsilon \max_{y_1 \in (y_1^*, \hat{y}_1)} \frac{1}{f(y_1|\boldsymbol{x})} \\
&\leq \frac{2\varepsilon}{\eta}.
\end{aligned}$$

$\qquad\square$

### C.2.1 Proof of Theorem 2

We can now finally combine all these results to prove Theorem 2.

*Proof of Theorem 2.* From proposition 1 we have that

$$\mathbb{P}\left(\left|\hat{h}(y_0, y_1|\boldsymbol{x}) - h^*(y_0, y_1|\boldsymbol{x})\right| \leq \varepsilon_h(n, \delta)\right) \geq 1 - \delta$$

with

$$\varepsilon_h(n, \delta) := \varepsilon_{T_n}(n, \delta/4) + \varepsilon_\alpha(n, \delta/4)\varepsilon_\beta(n, \delta/4) + \varepsilon_\gamma(n, \delta/4).$$

Now define $\{Y^{(i)}\}_{i=1}^n$ to be the sorted version of $\{Y_i\}_{i=1}^n$. Then by Proposition 6 we have

$$\mathbb{P}\left(\max_{j \in [n]_0} |\hat{h}(y_0, Y_1^{(i)}|\boldsymbol{x}) - h^*(y_0, Y_1^{(i)}|\boldsymbol{x})| \geq \varepsilon_{h-step}(n, \delta)\right) \leq \delta.$$

Plugging this into Proposition 9 with $\varepsilon_{step}(n, m, \delta)$ replaced by $\xi^{-1}\varepsilon_{\boldsymbol{w}}(n, \delta/n))$ and $\varepsilon_h(n, \delta)$ replaced with $\varepsilon_h(n, \delta/2)$ gives

$$\mathbb{P}\left(\sup_{y_1 \in \mathcal{Y}} \left|\tilde{h}(y_0, y_1|\boldsymbol{x}) - h^*(y_0, y_1|\boldsymbol{x})\right| \leq \varepsilon_h(n, \delta/(2n)) + \xi^{-1}\varepsilon_{\boldsymbol{w}}(n, \delta/n))\right) \geq 1 - \delta.$$

Now let $y_1^* := g^*(y_0|\boldsymbol{x})$. Then for any $y_1 \in \mathcal{Y}$ if $|y_1' - y_1^*| > s$ this implies that $|F_1(y_1'|\boldsymbol{x}) - F_1(y_1^*|\boldsymbol{x})| > s\eta$ by our lower bound on the density in $B_s(y_1^*)$. Hence by the contrapositive, if $|F_1(y_1'|\boldsymbol{x}) - F_1(y_1^*|\boldsymbol{x})| \leq s\eta$ then $|y_1' - y_1^*| < s$.

Now as

$$|F_1(y_1'|\boldsymbol{x}) - F_1(y_1^*|\boldsymbol{x})| \leq s\eta \Leftrightarrow y_1' \in F_1^{-1}(B_{s\eta}(F_1(y_1^*)))$$
$$\Leftrightarrow y_1' \in B_{s\eta}F_1^{-1}(B_{s\eta}(F_0(y_0)))$$

Hence if $n$ is sufficiently large so that $\varepsilon := \varepsilon_h(n, \delta/(2n)) + \xi^{-1}\varepsilon_{\boldsymbol{w}}(n, \delta/n)) \leq \eta s$ then we satisfy the bounded density condition of Theorem 10. Therefore we can plug our bound into theorem 10 gives

$$\mathbb{P}\left(|\hat{g}(y_0|\boldsymbol{x}) - g^*(y_0|\boldsymbol{x})| \leq 2\left(\eta^{-1}\varepsilon_h(n, \delta/(2n)) + \xi^{-1}\varepsilon_{\boldsymbol{w}}(n, \delta/n))\right)\right) \geq 1 - \delta.$$

$\square$

### C.3 Extension to expectation

**Proposition 11.** *Let $Y_0, D$ be RVs on $\mathcal{Y}, \mathcal{Z}^n$ respectively (with $D$ representing data used to fit a model and $Y_0$ representing the point where the model is fit). Now take $l(Y_0, D)$ to be a non-negative bounded loss so that $l(Y_0, D) < l_{\max}$ a.s. . Suppose that for any $\delta > 0$, for all $y \in \mathcal{Y}$*

$$\mathbb{P}(l(y, D) > \varepsilon(n, \delta)) < \delta$$

*Then for any $t, \delta_0 \in [0, 1]$ and $p, q \in [1, \infty]$ such that $1/p + 1/q = 1$*

$$\mathbb{P}\left(\mathbb{E}[l(Y_0, D)|D] \leq t^{1/q}\mathbb{E}[l(Y_0, D)^p|D]^{1/p} + \varepsilon(n, \delta_0)\right) \geq 1 - \frac{\delta_0}{t}.$$

*In particular if $l(y, D) < l_{\max}$ a.s. then taking $q = 1, p = \infty$ yields*

$$\mathbb{P}\left(\mathbb{E}[l(Y_0, D)|D] \leq tl_{\max} + \varepsilon(n, \delta_0)\right) \geq 1 - \frac{\delta_0}{t}.$$

*Proof.* First fix $t, \delta_0 \in [0, 1]$ We will first bound the probability that the number of $y$ which don't satisfy our bound isn't too large. We do this by defining the event

$$A := \{l(Y_0, D) > \varepsilon(n, \delta_0)\},$$

and now aim to bound the probability that $\mathbb{P}(A|D) > t$ (this probability is just w.r.t $Y_0$ treating $D$ as fixed.)

By Markov's inequality,

$$\mathbb{P}(\mathbb{P}(A|D) > t) \leq \frac{1}{t}\mathbb{E}[\mathbb{P}(A|D)]$$

$$= \frac{1}{t}\mathbb{E}[\mathbb{P}(A|Y_0)] \quad \text{by Fubini's Theorem}$$

$$< \frac{1}{t}\mathbb{E}[\delta_0] = \frac{\delta_0}{t}.$$

Now define $B := \{\mathbb{P}(A|D) \leq t\}$ so that $\mathbb{P}(B) = 1 - \frac{\delta_0}{t}$. Then under $B$

$$\mathbb{E}[l(Y_0, D)|D] = \mathbb{E}[\mathbb{1}_A l(Y_0, D)|D] + \mathbb{E}[l(Y_0, D)|A^c, D]\mathbb{P}(A^c|D)$$

$$\leq \mathbb{E}[\mathbb{1}_A l(Y_0, D)|D] + \varepsilon(n, \delta_0) \quad \text{by definitions of } A$$

$$\leq \mathbb{E}[\mathbb{1}_A|D]^{1/q}\mathbb{E}[l(Y_0, D)^p|D]^{1/p} + \varepsilon(n, \delta_0) \quad \text{by Holder's inequality}$$

$$\leq t^{1/q}\mathbb{E}[l(Y_0, D)^p|D]^{1/p}t + \varepsilon(n, \delta_0) \quad \text{As we are assuming } B.$$

$\square$

**Corollary 12.** *Assume that $F_1(y|\boldsymbol{x}) > \eta$ for all $y_1 \in \mathcal{Y}$. Then, for our fixed $\boldsymbol{x} \in \mathcal{X}$ and $\delta \in (0, e^{-1})$, w.p. at least,*

$$1 - \frac{\delta\operatorname{diam}(\mathcal{Y})}{2\left(\eta^{-1}\varepsilon_h(n, \delta/2n) + \xi^{-1}\varepsilon_{\boldsymbol{w}}(n, \delta/n))\right)},$$

$$\mathbb{E}\left[|\hat{g}(Y|\boldsymbol{x}) - g^*(Y|\boldsymbol{x})| \,\Big|\, A = 0, \hat{g}\right] \leq 4\left(\eta^{-1}\varepsilon_h(n, \delta/n) + 2\xi^{-1}\varepsilon_{\boldsymbol{w}}(n, \delta/n)\right)$$

*where* $\varepsilon_h(\delta, n) := \varepsilon_{T_n}(n, \delta/4) + \varepsilon_\alpha(n, \delta/4)\varepsilon_\beta(n, \delta/4) + \varepsilon_\gamma(n, \delta/4).$

*In particular for* $2\left(\eta^{-1}\varepsilon_h(n, \delta/n) + 2\xi^{-1}\varepsilon_{\boldsymbol{w}}(n, \delta/n)\right) \leq \log(e_1(n)/\delta)^a/e_2(n)$. *For $\delta < 1/e$ we have that w.p. at least $1 - \delta$*

$$\mathbb{E}\left[|\hat{g}(Y|\boldsymbol{x}) - g^*(Y|\boldsymbol{x})| \,\Big|\, A = 0, \hat{g}\right] \leq \log(2\operatorname{diam}(\mathcal{Y})e_2(n)e_1(n)/\delta)^a/e_2(n)$$

*Proof.* Plugging the result of theorem 2 into proposition 11, noting that $|\hat{g}(y_0|\boldsymbol{x}) - g^*(y|\boldsymbol{x})| \leq \operatorname{diam}(\mathcal{Y})$ and taking

$$t = \frac{2}{\operatorname{diam}(\mathcal{Y})}\left(\eta^{-1}\varepsilon_h(n, \delta/(2n)) + \xi^{-1}\varepsilon_{\boldsymbol{w}}(n, \delta/n)\right)$$

yields that w.p. at least

$$1 - \frac{\delta\operatorname{diam}(\mathcal{Y})}{2\left(\eta^{-1}\varepsilon_h(n, \delta/n) + \xi^{-1}\varepsilon_{\boldsymbol{w}}(n, \delta/n))\right)},$$

$$\mathbb{E}\left[|\hat{g}(Y|\boldsymbol{x}) - g^*(Y|\boldsymbol{x})| \,\Big|\, A = 0, \hat{g}\right] \leq 4\left(\eta^{-1}\varepsilon_h(n, \delta/n) + 2\xi^{-1}\varepsilon_{\boldsymbol{w}}(n, \delta/n)\right).$$

Following from this if $2\left(\eta^{-1}\varepsilon_h(n, \delta/n) + 2\xi^{-1}\varepsilon_{\boldsymbol{w}}(n, \delta/n)\right) \leq \log(e_1(n)/\delta)^\alpha/e_2(n)$. Then we have

$$\mathbb{P}\left(\mathbb{E}\left[|\hat{g}(Y|\boldsymbol{x}) - g^*(Y|\boldsymbol{x})| \,\Big|\, A = 0, \hat{g}\right] \geq 2\log(e_1(n)/\delta)^a/e_2(n)\right) \leq \frac{\delta\operatorname{diam}(\mathcal{Y})e_2(n)}{\log(e_1(n)/\delta)^a}$$

for $\delta < 1/e$ we have

$$\delta < 1/e \Rightarrow \delta < \exp\left\{-(\frac{1}{2})^{1/a}\right\}$$

$$\Leftrightarrow \log(1/\delta)^a > \frac{1}{2}$$

$$\Rightarrow \log(e_1(n)/\delta)^a > \frac{1}{2}$$

$$\Rightarrow 2\delta\log(e_1(n)/\delta)^a > \delta$$

$$\Leftrightarrow 2\delta > \frac{\delta}{\log(e_1(n)/\delta)^a}.$$

Hence

$$\mathbb{P}\left(\mathbb{E}\left[|\hat{g}(Y|\boldsymbol{x}) - g^*(Y|\boldsymbol{x})| \ \Big| A = 0, \hat{g}\right] \geq 2\log(e_1(n)/\delta)^a/e_2(n)\right) \leq 2\delta\,\mathrm{diam}(\mathcal{Y})e_2(n).$$

Finally $\delta' = 2\delta\,\mathrm{diam}(\mathcal{Y})e_2(n)$ gives

$$\mathbb{P}\left(\mathbb{E}\left[|\hat{g}(Y|\boldsymbol{x}) - g^*(Y|\boldsymbol{x})| \ \Big| A = 0, \hat{g}\right] \geq 2\log(2\,\mathrm{diam}(\mathcal{Y})e_1(n)e_2(n)/\delta')^a/e_2(n)\right) \leq \delta'.$$

$\square$

### C.4   Application and justification of NW estimation with box kernel

We aim to show that the box kernel satisfies some of our conditions. Specifically conditions 2 & 3 from Proposition 1. We first start by bounding the step size.

**Proposition 13** (Effective sample size). *Suppose that for our $\boldsymbol{x} \in \mathcal{X}$ there exists $C_0, r_0 > 0$ such that for any $r \in (0, r_0)$*

$$\mathbb{P}(X \in B_r(\boldsymbol{x})) \geq C_0 r^d.$$

*Now for $r \in (0, r_0)$ take our kernel to be $k_r(\boldsymbol{x}, \boldsymbol{x}') := \mathbb{1}\{\|\boldsymbol{x} - \boldsymbol{x}'\| \leq r\}$. Then w.p. at least $1 - \delta$*

$$\sum_{j \in \mathcal{J}} \mathbb{1}\{X_j \in B_r(\boldsymbol{x})\} \geq \left(nC_0 r^d - \sqrt{2nC_0 r^d \log(1/\delta)} - \log(1/\delta)/3\right)^{-1}.$$

*Proof.* We have $\mathbb{1}\{X_j \in B_r(\boldsymbol{x})\} \leq 1$ and $\mathbb{E}[\mathbb{1}\{X_j \in B_r(\boldsymbol{x})\}^2] = \mathbb{E}[\mathbb{1}\{X_j \in B_r(\boldsymbol{x})\}^2] = \mathbb{P}(X_j \in B_r(\boldsymbol{x})) = C_0 r^d$. Therefore by one sided Bernstein's inequality with $\varepsilon = \log(1/\delta)$ we get

$$\mathbb{P}\left(\sum_{j \in \mathcal{J}} \mathbb{1}\{X_j \in B_r(\boldsymbol{x})\} \leq nC_0 r^d - \sqrt{2nC_0 r^d \log(1/\delta)} - \frac{1}{3}\log(1/\delta)\right) \leq \delta.$$

This gives our desired result. $\square$

Note that $\sum_{j \in \mathcal{J}} \mathbb{1}\{X_j \in B_r(\boldsymbol{x})\}$ is also the effective sample size of our estimation as it is the number of samples used in the average.

Now that we have the effective sample size result in terms of our kernel radius $r$, we need to obtain the optimal rate of decay of $r$ for our estimation.

**Proposition 14** (NW estimation with box kernel). *let $\hat{m}_f(\boldsymbol{x})$ be the NW estimation of $m_f(\boldsymbol{x}) := \mathbb{E}[f(Z)|X = \boldsymbol{x}]$ using IID copies $(Z_i)_{i=1}^n$ of $Z$. and assume $|f(Z)| \leq M$. For a fixed $r \in (0, r)$, use kernel $k_r$ as defined above and suppose the same assumptions hold. Suppose that $m_f(\boldsymbol{x})$ is $\alpha$ smooth for $\alpha \leq 1$ (i.e. $\alpha$-Holder continuous.) Then for sufficiently large $n$ depending on $C_0, \alpha$ and $\delta \leq 2e^{-1}$ w.p. at least $1 - \delta$,*

$$|\hat{m}_f(\boldsymbol{x}) - m_f(\boldsymbol{x})| \leq \frac{2M+1}{C_0}\log(2/\delta)n^{-\frac{1}{2+d/\alpha}}.$$

*Proof.* With our kernel define $\mathcal{I}_r := \{i \in [n]\,|\,k_r(\boldsymbol{x}, X_i) = 1\}$ and $n_r = |I_r|$. Now define the event

$$E_n := \{n_r \geq C_0 n r^d - \sqrt{2C_0 r^d n \log(2/\delta)} - \log(2/\delta)/3\}.$$

Then by Proposition 13 this event occur w.p. at least $1-\delta/2$. Now if we define $\varepsilon_i := f(Z_i) - m_f(X_i)$ then we have $\mathbb{E}[\varepsilon_i|X_i] = 0$. Also, we have

$$|\hat{m}_f(\boldsymbol{x}) - m_f(\boldsymbol{x})| = \left| \frac{1}{n_r} \sum_{i \in \mathcal{I}_r} f(Z_i) \; - \; m_f(\boldsymbol{x}) \right|$$

$$= \left| \frac{1}{n_r} \sum_{i \in \mathcal{I}_r} m_f(X_i) + \varepsilon_i \; - \; m_f(\boldsymbol{x}) \right|$$

$$\leq \frac{1}{n_r} \sum_{i \in \mathcal{I}_r} |m_f(X_i) - m_f(\boldsymbol{x})| + \left| \frac{1}{n_r} \sum_{i \in \mathcal{I}_r} \varepsilon_i \right|$$

$$\leq r^\alpha + \left| \frac{1}{n_r} \sum_{i \in \mathcal{I}_r} \varepsilon_i \right|.$$

With the final equality coming by our smoothness condition and definition of $\mathcal{I}_r$. Now by Hoeffding bounds we have that

$$\mathbb{P}\left( \left| \sum_{i \in \mathcal{I}_r} \varepsilon_i \right| \geq \sqrt{\frac{2\log(4/\delta)M^2}{n_r}} \right) \leq \frac{\delta}{2}.$$

Hence by combining this event and $E_n$ then we have that with probability at least $1 - \delta$

$$|\hat{m}_f(\boldsymbol{x}) - m_f(\boldsymbol{x})| \leq r^\alpha + \sqrt{\frac{2\log(2/\delta)M^2}{n_r}}$$

$$\leq r^\alpha + \sqrt{\frac{2\log(2/\delta)M^2}{C_0 n r^d - \sqrt{2C_0 n r^d \log(2/\delta)} - \log(2/\delta)/3}}.$$

Now for sufficiently large $n$

$$C_0 n r^d - \sqrt{2C_0 n r^d \log(2/\delta)} - \log(2/\delta)/3 \geq \frac{C_0 n r^d}{2\log(2/\delta)}.$$

This in turn gives

$$|\hat{m}_f(\boldsymbol{x}) - m_f(\boldsymbol{x})| \leq r^\alpha + \frac{2\log(2/\delta)M}{\sqrt{C_0 n r^d}}.$$

Then the optimal choice of $r$ is such that

$$r^\alpha = \frac{1}{\sqrt{n r^d}}$$

$$\Leftrightarrow r^{\alpha + \frac{d}{2}} = n^{-\frac{1}{2}}$$

$$\Leftrightarrow r = n^{-\frac{1}{2\alpha + d}}$$

$$\Leftrightarrow r^\alpha = n^{-\frac{1}{2 + d/\alpha}} = \frac{1}{\sqrt{n r^d}}.$$

Hence plugging this in we get that for sufficiently large $n$ depending on $C_0, \alpha$ we have that for $\delta \leq 2e^{-1}$ w.p. at least $1 - \delta$

$$|\hat{m}_f(\boldsymbol{x}) - m_f(\boldsymbol{x})| \leq \frac{2M + 1}{C_0} \log(2/\delta) n^{-\frac{1}{2 + d/\alpha}}.$$

$\square$

**Proposition 15** (Final weight decay). *Suppose that for our $\boldsymbol{x} \in \mathcal{X}$ there exists $C_0, r_0 > 0$ such that for any $r \in (0, r)$*

$$\mathbb{P}(X \in B_r(\boldsymbol{x})) \geq C_0 r^d.$$

Suppose that we are performing NW estimation of an $\alpha$ smooth function at points $\boldsymbol{x} \in \mathcal{X}$ using kernel $k_{r_n}(\boldsymbol{x}, \boldsymbol{x}') := \mathbb{1}\{\|\boldsymbol{x} - \boldsymbol{x}'\| \le r\}$ with $r_n$ decaying optimally. Now define

$$w_j := \frac{k_{r_n}(\boldsymbol{x}, X_j)}{\sum_{j' \in \mathcal{J}} k_{r_n}(\boldsymbol{x}, X_{j'})}$$

the weight of the $j^{th}$ component. Then with probability at least $1 - \delta$

$$\max_{j \in \mathcal{J}} w_j \le = \frac{\log(2/\delta)}{C_0} n^{\frac{-2}{2+d/\alpha}}.$$

*Proof.* We have

$$\max_{j \in \mathcal{J}} w_j \le \frac{1}{\sum_{j \in \mathcal{J}} k(\boldsymbol{x}, X_j)}$$

$$= \frac{1}{\sum_{j \in \mathcal{J}} \mathbb{1}\{X_j \in B_r(\boldsymbol{x})\}}.$$

Hence by proposition 13 we have w.p. at least $1 - \delta$

$$\max_{j \in \mathcal{J}} w_j \le \left( C_0 n r^d - \sqrt{2 C_0 n r^d \log(1/\delta)} - \log(1/\delta)/3 \right)^{-1}.$$

We know from Proposition 14 that the optimal radius decay gives $\frac{1}{n r^d} = n^{-\frac{2}{2+d/\gamma}}$. Plugging this into our result gives that

$$\max_{j \in \mathcal{J}} w_j \le \left( C_0 n^{\frac{2}{2+d/\gamma}} - \sqrt{2 C_0 \log(1/\delta)} n^{\frac{1}{2+d/\gamma}} - \log(1/\delta)/3 \right)^{-1}$$

$$\le \frac{\log(1/\delta)}{C_0 n^{\frac{2}{2+d/\gamma}}} \quad \text{For sufficiently large } n \text{ depending on } \gamma, C_0.$$

$\square$

Note that for a $\gamma$ smooth function the MSE is $C^{-1} n^{-\frac{1}{2+d/\gamma}}$. Hence the box kernel comfortably satisfies our weight decay condition 2 in Assumptions 2.

Additionally now if we have an $\alpha$ smooth function and assume that for any $\boldsymbol{x}' \in \mathcal{X}$ there exists $C_0(\boldsymbol{x}')$ such that for all $r \in (0, r_0) \, \mathbb{P}(X \in B_r(\boldsymbol{x}') \ge C_0 r^d$. Then we have that w.p. at least $1 - \delta$,

$$w_{F_a;i}(X, X^{\mathcal{I}}) \le \frac{\log(2/\delta)}{C_0(X)} n^{-\frac{2}{2+d/\alpha}}.$$

Thus if we assume that there exists $C'' > 0$ such that w.p. at least $1 - \delta$, $\frac{1}{C_0(X)} \le C'' \sqrt{\log(1/\delta)}$ then we get that w.p. at least $1 - \delta$

$$w_{F_a;i}(X, X^{\mathcal{I}}) \le C'' \log^{3/2}(3/\delta) n^{-\frac{2}{2+d/\alpha}}.$$

This then gives our condition 2 in Assumptions 2.

We now try to bound $\frac{\sum_{j \in \mathcal{J}} w_j^2 \sigma(X_j)}{\mathbb{E}\left[ \sum_{j \in \mathcal{J}} w_j^2 \sigma(X_j) \right]}$.

**Corollary 16** (Accuracy under NW estimation with box kernel). *Suppose that our linear smoother is NW estimation with the box kernel and optimally decaying radius additionally assume that:*

- $\varepsilon_{\hat{\varphi}}(n, \delta) = e_{\hat{\varphi}}(n) \sqrt{\log(1/\delta)}$ with $e_1 = o(1)$.

- $\varepsilon_\alpha(n, \delta) = C_\alpha \sqrt{\log(1/\delta)} n^{-\frac{1}{2+d/\alpha}}$.

- $\varepsilon_\beta(n, \delta) = C_\beta \sqrt{\log(1/\delta)} n^{-\frac{1}{2+d/\beta}}$.

- $\beta > \frac{d}{2(1+d/\gamma)}$. *Then for any $\delta$, for sufficiently large $n$ the following events each separately hold w.p. at least $1 - \delta$,*

$$\left|\hat{h}(y_0, y_1|\boldsymbol{x}) - h^*(y_0, y_1|\boldsymbol{x})\right| \leq C_h \frac{\log^{3/2}(1/\delta)}{e_h(n)}$$

$$|\hat{g}(y|\boldsymbol{x}) - g^*(y|\boldsymbol{x})| \leq \frac{C_g}{\eta\xi} \frac{\log^{3/2}(n/\delta)}{e_h(n)}$$

$$\mathbb{E}\left[|\hat{g}(Y|\boldsymbol{x}) - g^*(Y|\boldsymbol{x})| \;\Big|\; A = 0, \hat{g}\right] \leq \frac{C_{g,2} \operatorname{diam}(\mathcal{Y})}{\xi\eta} \frac{\log^{3/2}(e_2(n)n/\delta)}{e_2(n)}$$

*with $C_h, C_g, C_{g,2}$ depending on $C_\alpha, C_\beta, C', C''$ (where $C', C''$ are the constants define in Proposition 15.*

*Proof.* For any NW estimator we have that $\|\boldsymbol{w}\|_1, \|\boldsymbol{w}\|_2 \leq 1$ a.s. meaning we can take $\varepsilon_{\boldsymbol{w}}(n, \delta) \equiv 1$. We also have that $\varepsilon_\gamma(n, \delta) = C_\gamma \log(1/\delta) n^{-\frac{1}{1+d/\gamma}}$. Hence $\varepsilon_{T_n} = \sqrt{2\varepsilon_\delta(n)} e_{\hat{\varphi}}(n) \sqrt{\log(1/\delta)}$. This then gives

$$\left|\hat{h}(y_0, y_1|\boldsymbol{x}) - h^*(y_0, y_1|\boldsymbol{x})\right| \leq C_\gamma \log^{1/2}(7/\delta) n^{-\frac{1}{2+d/\gamma}} + C_\alpha C_\beta \log(7/\delta) n^{-\left(\frac{1}{2+d/\alpha} + \frac{1}{2+d/\beta}\right)}$$

$$+ \sqrt{2\varepsilon_{\delta/4}(n)} e_{\hat{\varphi}}(n) \sqrt{\log(7/\delta)}$$

$$\leq C_h \log(1/\delta) n^{-\frac{1}{2+d/\gamma}} + \log(1/\delta) n^{-\left(\frac{1}{2+d/\alpha} + \frac{1}{2+d/\beta}\right)}$$

$$\leq C_h \frac{\log(1/\delta)}{e_h(n)}$$

for $\delta < e^{-1}$ and $C_h = 7(C_\gamma + C_\alpha C_\beta + \sqrt{2})$ giving our first result.

Using our weight decay results for NW estimation and plugging into Proposition 6 we get that

$$\varepsilon_{h-step} = C'' \xi^{-1} \frac{\log^{3/2}}{e_h(n)}(n/\delta).$$

Hence plugging into Theorem 2 gives

$$|\hat{g}(y|\boldsymbol{x}) - g^*(y|\boldsymbol{x})| \leq C_h \sqrt{2} \eta^{-1} \frac{\log(n/\delta)}{e_h(n)} + 4C'' \xi^{-1} \log^{3/2}(n/\delta) n^{-\frac{1}{2+d/\gamma}}$$

$$\leq \frac{C_g}{\xi\eta} \frac{\log^{3/2}(n/\delta)}{e_h(n)}$$

with $C_g = C_h \sqrt{2} + 4C''$ giving our second result. Finally by Corollary 12 for $\delta < e^{-1}$ w.p. at least $1 - \delta$,

$$\mathbb{E}\left[|\hat{g}(Y|\boldsymbol{x}) - g^*(Y|\boldsymbol{x})| \;\Big|\; A = 0, \hat{g}\right] \leq \frac{C_g \operatorname{diam}(\mathcal{Y})}{\xi\eta} \frac{\log^{3/2}(ne_h(n)/\delta)}{e_h(n)}.$$

$\square$

Note that both $\varepsilon_{T_n}$ and $\varepsilon_{h-step}$ are actually both $o(\varepsilon_\gamma(n, c))$, thus if we were allowed to take $n$ sufficiently large depending upon $\delta$ then we would have all $\log^{3/2}$ terms replaced with $\log$ terms and the dependence on $C', C''$ removed.

