# OpenReview forum: "Conditional Outcome Equivalence: A Quantile Alternative to CATE"
_NeurIPS.cc/2024/Conference — NeurIPS 2024 poster_

### Official Review · Reviewer_zMNr · 2024-07-09

**Soundness:** 3
**Presentation:** 3
**Contribution:** 3
**Rating:** 6
**Confidence:** 3

**Summary:**

The paper introduces a new estimator to obtain conditional quantiles of a treatment effect and argues that it is superior to the pre-existing "conditional quantile treatment effect" (CQTE) estimator. The main difference is that accurate CQTE estimates require accurate conditional quantile models even though the latter can be hard to fit, while the introduced CQC estimator is more robust to the CDF estimators it relies on.

**Strengths:**

Introduces the doubly robust property to estimating conditional quantile treatment effects, a property that many CATE estimators already have.

Detailed theory work supporting the claim of robustness of CQC to its component estimators.

Exploits existing work on doubly robust CATE estimation by reframing CCDF estimation as a CATE-like problem.

**Weaknesses:**

In 4.1 you test against estimating the CCDFs separately and taking their difference. Do you mean estimating the inverse CDFs / quantile functions and taking their difference i.e. CQTE? If not, what exactly is the procedure here and why not compare against the difference of two quantile estimators.

CCDFs are arguably harder to estimate than quantile functions in my experience (they require the additional 'feature' y, have to work with classification losses instead of regression losses, and are more prone to monotonicity violations). So even if the final estimator here is more robust to bad CCDF estimation I'm not convinced it will necessarily be better in practice to the CQTE approach of fitting quantile functions.

This is especially true with modern day neural network based quantile models. NW estimators are unlikely to be as good as those on larger / higher dimension datasets, and it's not clear how the theory would hold in cases like that.

Isotonization can be risky and create large flat zones for $\hat{g}$. Did you notice any in your experiments? Are there other ways to get monotonicity here, similar to the various strategies to get monotonicity in multi-quantile regression?

**Questions:**

See above

**Limitations:**

Yes

---

> ### Author Rebuttal · Authors · 2024-08-07
>
> We thank the reviewer for their detailed and salient comments and aim to address them now.
>  ___
> # Clarification on the separate estimator
> We appreciate that we did not provide enough information on this separate estimator and will ensure to include its explicit form in the paper.
> We do not quite take the inverse CDFs and subtract them, as that would provide an estimate of the CQTE, which is different from our CQC estimand.
>
> Our separate estimator can be thought of in two ways. For both we have estimates $\hat F_{Y|X,1},\hat F_{Y|X,0}$ of the CCDFs. We either use:
>
> 1. $\hat g=\hat F_{Y|X,1}^{-1}(\hat F_{Y|X,0}(y_0|x)|x)$
> 2. $\hat g=\hat h^{-1}(y_0,0|x)$ with $\hat h(y_0,y_1|x)=\hat F_{Y|X,1}(y_1|x)-\hat F_{Y|X,0}(y_0|x)$ and the inverse of $\hat h$ being w.r.t. the second argument.
>
> Both methods are actually equivalent. The first method may be more intuitive, while the second method involves estimating $\hat h$ using the classic plug-in estimator (which is equivalent to the T-learner in CATE) and then inverting to obtain $\hat g$, as we also do in our DR procedure.
>
> Returning to the separate estimator you mentioned (i.e., taking the difference of the two inverse CCDFs), if we were to compute this and then transform it using our estimated CCDFs to obtain the CQC (as detailed in the transformation between the CQC and CQTE in Section 2.2), it would be numerically very close to the two approaches described above. Any differences would arise from the fact that the inverses of step functions are not true inverses but generalised inverses.
> ___
> # Estimating the CCDFs vs the quantiles
> We don't see personally see evidence to suggest that CCDFs are more difficult to estimate than quantile functions. In fact, a common non-parametric kernel approach for estimating quantile functions involves first estimating the CCDFs and then inverting this estimate. Additionally, due to the monotonicity of the true CCDF, the price we pay for estimating over all $y$ rather than a single point is very small. This is reflected in our theoretical results, which show only a $\log(n)$ slower convergence rate compared to the equivalent results for CATE estimation.
>
> Regarding the ensuring of monotonicity in our estimate, we use the method from Yang and Barber (2019). This approach allows us to initially obtain a non-monotonic estimator and then improve it by monotonically projecting. Therefore, we do not need to be overly concerned about monotonicity in our initial estimator.
> ___
> # NW estimation and issues with high dimensionality
> While we agree that NW estimation may be impractical for high-dimensional problems, our theory applies to the case of linear smoothers, which is a broad class that includes generalised random forests, k-NN regression, and methods often associated with higher-dimensional data, such as kernel ridge regression (see the global response for references and more detail). We use NW estimation as an illustrative regression technique since this paper focuses on the estimation of CQC rather than on the regression techniques themselves. Additionally, our pseudo-outcome regression could be readily extended to incorporate neural networks or other high-dimensional regression techniques.
> ___
> # Issues with isotonization
> Thank you for flagging this issue. We agree that isotonization can cause problems when applied to functions that significantly violate the monotonicity assumption. In practice, however, we find that our approach produces functions that are already very close to isotonic, meaning the projection has minimal effect. This is illustrated in Figure 1, where we compare the method with isotonic projection to the method without isotonic projection, taking the inverse to be the first point where it exceeds zero. As shown, the results are indistinguishable (red and purple lines).
>
> We focused specifically on a post-processing approach because it allows us to preserve the accuracy of our original, unmodified estimator. While other methods could be used to guarantee monotonicity, they would be incompatible with our theoretical results. Given that our monotonicity violations are relatively small, post-processing feels the most logical approach.
> ___
> # References
> Yang, F. and Barber, R. F. (2019). Contraction and uniform convergence of isotonic regression. Electronic
> Journal of Statistics, 13(1):646 – 677. Publisher: Institute of Mathematical Statistics and Bernoulli Society.

---

> > ### Comment · Reviewer_zMNr · 2024-08-13
> >
> > Thanks for the response! Appreciate the extra info and I feel more confident in my already positive score.

---

### Official Review · Reviewer_CQR9 · 2024-07-10

**Soundness:** 3
**Presentation:** 2
**Contribution:** 3
**Rating:** 6
**Confidence:** 4

**Summary:**

The paper studies an important problem -- treatment effects often have distributional effects and considering conditional mean outcomes is often not the correct objective for evaluating prescriptive interventions. One popular approach to consider heterogeneous effects across a population which allows researchers to determine treatment effects at different quantiles of the study distribution to determine how treatments might effect the worst-off populations. Unfortunately, as the authors note obtaining quantile treatment effects often relies strict smoothness assumptions on the marginal quantile functions and lacks the double robustness properties of CATE estimators. To this end, the authors propose a new estimand to evaluate heterogenous treatment effects that they show too be doubly robust and apply this estimator to both simulational and real-world data.

**Strengths:**

- The paper provides a clean mathematical framework for their estimator and proves useful finite sample properties that should be theoretically convincing for users
- Example 1 is a nice simulational example to showcase why typical conditional quantile estimators may provide instable results when the author's CQC estimator is more robust
- Proposition 1 provides the double robustness result which has been a useful benefit in finite sample analyses and helps to make the case for using this estimator
- Numerical experiments in section 4 help to show case the benefits of this procedure. In particular the second graph in Figure 3 provides good evidence for using this methodology.
- Finally, the application to two real-world data sets is helpful to showcase the viability of this methodology for practioners.

**Weaknesses:**

- I understand that the CQTE can be written as a function of $\Delta^*$ but this does not help me to understand how I should interpret this quantity in relation to the typical CQTE. The authors indicate that this it is a "rephrasing" but and provide the mapping. but its not obvious how this translates in actual examples. Since the authors are proposing a non-standard estimand they should provide more details linking this to the standard analysis. One place where this would be very beneficial is in the interpretation of the real-world data sets. I consider this to be the main weakness of the work because without a clear interpretation for the estimand practioners will not adopt this methodology. Without this I doubt the impact of the paper beyound showcasing a technical deficiency in the CQTE estimand due to smoothness requirements.
- I found the algorithms a little difficult to understand especially because of the reindexing where you duplicate the data. This seems purely notational and seems like you could simplify the notation here.
- The method does not require smoothness of the marginal quantiles, however h must be Holder-smooth. Certainly in many cases smoothness of the marginal may effect smoothness of h. Since the reason for introducing this method is non-smoothness of the marginal quantiles it might be useful to expand on the class of problems where this is an easier problem beyound the example given in Example 1. How does this interact with smoothness of g?
- To that end, could you also provide real-world examples where you might expect the marginals to be poorly behaved but the CQC to be well be haved?
- Finally, in the the final paper in paper it would be good to have the CQTE estimates in the real world examples so you can draw a contrast between the methods.

**Questions:**

In remark 3 you mention you use a cross-fitting method to improve stability, why is this the case? Duplicating the data yields non-independence?
How do i interpret the earnings for groups with the highest/lowest benefits in Figure 4?

**Limitations:**

The authors have addressed some limitations of their work but have not acknowledged possible negative societal impact. From my perspective, work that emphasizes the importance of considering distributional effects is generally societally beneficial. The main limitation that the authors address are primarily on the interplay between the assumption on h and if they could be relaxed to assumptions on g. However, there are several other assumptions and limitations that should be made clear in this work. The authors should potentially consider including limitations in their analysis of the two real-world data sets which might give practitioners a better understanding of how to use this method. Examples of what could be added are given in the weakness section especially with regards to interpretability of results.

---

> ### Author Rebuttal · Authors · 2024-08-07
>
> We thank the reviewer for their detailed and salient comments and aim to address them now.
> ___
> # Interpreting the CQTE
> We appreciate how crucial it is to properly justify and explain a new estimate and will make sure to provide more intuition and interpretation for our estimator in the final document.
>
> As a brief insight, we argue that the CQC is inherently a more intuitive object overall than the CQTE, as it more naturally characterises the effect of a treatment. To compare with CATE, consider the canonical example in the potential outcome framework where we assume $Y_1=\phi(X)+Y_0$ for some $\phi$. In this case, $\phi(X)$ is the CATE, and so a smooth CATE is given by a smooth $\phi$. The CQC generalises this notion beyond additive treatment effects to the case $Y_1=\phi(Y_0,X)$, where $\phi$ is any function that increases in $Y_0$ (the CATE case is a subset of this). If we want to express this in the CATE form, we can write it as $Y_1=\Delta^*(Y_0,X)+Y_0$.
>
> A simple example of this might be a treatment that halves the response for any individual (e.g., a blood pressure medication that decreases an individual's blood pressure by 50\%), in which case the CQC is $g^*(Y,X)=0.5Y$. Here, the CQC is constant with respect to the covariates, whereas the form of the CQTE will depend on $Y|X,A=0$, which may lead to a complicated dependence on $X$, hampering its interpretability (as will also be the case for the CATE).
>
> From another perspective, we frame our approach as a conditional QQ-plot. QQ-plots are already an established tool for comparing two unconditional 1-dimensional distributions. Our CQC function provides the QQ-plot conditioned on each possible value of our covariates, allowing us to apply this QQ-plot tool at the individual patient level, which is central to HTE studies.
>
> Overall, we see our approach as the most intuitive way of comparing the treated and untreated response across the entire distributions.
> ___
> # Algorithm clarity
> We appreciate that the reindexing in our algorithms adds complexity, making it more difficult to understand. We will find a more intuitive way to present this sample-splitting concept and potentially introduce it textually before the algorithm to avoid notation overload.
> ___
> # Smoothness of $h$ vs smoothness of $g$ and CCDFs
> We know that the smoothness of $h^*$ is a weaker assumption than the smoothness of the marginal distributions. For example, if $Y|X,A=0$ are both uniformly distributed with a fixed width (but with a non-smoothly varying centre with respect to $X$), and $Y_1=\phi(Y_0,X)$, then the CCDFs will be non-smooth, but the differences will be smooth.
>
> We believe that the smoothness of $h^*$ and $g^*$ are technically distinct concepts (with neither implying the other). However, in most practical examples (as illustrated above), when $h^*$ is smooth, so is $g^*$. Overall, we acknowledge that the smoothness of $h^*$ is less interpretable than the smoothness of $g^*$, and we consider extending to smooth $g^*$ as a goal for future work, as mentioned in our Limitations section.
> ___
> # Including CQTE in real world examples
> Thank you for the suggestion. We have added CQTE plots in the PDF, as shown in Figure 3, and will use this as an opportunity to highlight the relationship between the CQC and the CQTE in our paper. We will ensure to discuss the benefits and drawbacks of each approach with respect to interpretability here as well.
>
> In these plots, we can see that the interpretation of the CQTE is less obvious. This is because, for each covariate value, the quantile value corresponds to a different untreated response. We believe this makes direct comparisons between different covariate values less intuitive.
> ___
> # Questions
> > In remark 3 you mention you use a cross-fitting method to improve stability, why is
> this the case? Duplicating the data yields non-independence?
>
> In this context, improving stability means minimising the impact of our data split on the estimator. Cross-fitting will also maximise sample utilisation which we would expect to improve the estimation overall. Cross-fitting maximises sample utilisation and is therefore also expected to improve overall estimation accuracy. Using the same data for both inner and outer regressions would lead to dependence between the estimation of the outcome and the nuisance parameters, which would affect the theoretical results.
>
> > How do i interpret the earnings for groups with the highest/lowest benefits in Figure 4?
>
> By groups I assume you are referring to covariate groups. For the covariates, the groups with the most significant impact appear to be the youngest (those close to age 23), and individuals aged 30-36.
>
> For the youngest group, there is overall improvement, but most of this improvement occurs at the highest earnings. This suggests that the intervention increases the pay disparity between individuals, likely because those already earning more benefit the most. However, as with the CQTE, we cannot make strict assertions about individual-level impacts.
>
> In the 30-36 age group, there is a more even increase in incomes across the board. This indicates that the intervention shifts the entire income distribution up for this particular age group.
>
> Overall, those with the least benefit seem to be individuals with low non-intervention earnings and those aged 38-40. They experience minimal or no benefit from the intervention, with the value of $\Delta$ being close to 0.

---

### Official Review · Reviewer_nAy3 · 2024-07-11

**Soundness:** 3
**Presentation:** 2
**Contribution:** 2
**Rating:** 6
**Confidence:** 5

**Summary:**

The authors propose the Conditional Quantile Comparator (CQC), a function that maps an outcome from the control group in a binary treatment setting to an outcome in the treatment group, such that they represent the same conditional quantiles in their respective distributions. The estimation procedure consists of two stages: first, estimating a conditional contrasting function that examines the difference between conditional CDFs, followed by isotonic regression. The contrasting function is estimated using pseudo-outcome regression, which confers the algorithm with double-robust properties in finite samples. This estimator addresses some of the shortcomings of the Conditional Quantile Treatment Effect (CQTE), which is not robust to errors in the quantiles. The CQC is supported by a theoretical analysis of finite sample convergence rates under smoothness assumptions, as well as by empirical simulations.

**Strengths:**

The authors propose a novel quantile-based treatment effect estimation procedure tailored for skewed outcome distributions. The double-robustness properties in the first stage of the algorithm are desirable as they improve the rate dependence on the nuisance functions ( propensity and local CDF estimates). The paper is clearly written, with sound theoretical results and promising empirical evidence.

**Weaknesses:**

* The motivation for the estimator is somewhat weak.  Estimators for CQTEs with double-robustness properties have been already proposed (see [1, 2]). These estimators have a second order dependence on the rate of quantile estimation, as desired. The authors should contextualize their work with respect of exiting literature (which seems to overall be missing in the paper) and compare their algorithm with the existing techniques.
* Moreover, the pseudo-outcome  estimation technique in [1] is somewhat simpler (see their Appendix B) since they only require one pseudo-outcome regression. There seem to be some tradeoffs, e.g. the CQC requires a two stage procedure that includes a prost processing step (the isotonic regression), as well as for Algorithm 1 to be run several times for each value of $y_1$ considered (because the conditional CDFs need to be estimated at each value). On the other hand, the robust CQTE algorithm from [1] require estimating the conditional PDF at several points. Overall, the CQC seems to be a more computationally (and statistically) intensive estimation procedure and the authors should at least discuss the tradeoffs.
* Furthermore,  [1] provides rates for more general classes a functions outside of the linear smoothers framework in [3]. I suggest the authors try to provide similar guarantees in order to generalize their results beyond linear smoothers which, to the best of my knowledge, are not often used in practice. This would be particularly useful since the rate results in section 3.4 require all nuisances to be estimated using linear smoothers.
* Empirical evaluations should also include these existing methods.

Overall, I tend towards soft rejection. This estimator might be useful in its own right, but the authors should contextualize and compare with existing work. I am willing to update my score upon further discussion.

[1] Kallus, Nathan, and Miruna Oprescu. "Robust and agnostic learning of conditional distributional treatment effects." International Conference on Artificial Intelligence and Statistics. PMLR, 2023.

[2] Leqi, Liu, and Edward H. Kennedy. "Median optimal treatment regimes." arXiv preprint arXiv:2103.01802 (2021).

[3] Kennedy, Edward H. "Towards optimal doubly robust estimation of heterogeneous causal effects." Electronic Journal of Statistics 17.2 (2023): 3008-3049.

**Questions:**

See weaknesses section.

**Limitations:**

The authors have adequately adressed the limitations of their work.

---

> ### Author Rebuttal · Authors · 2024-08-07
>
> We thank the reviewer for their detailed and salient comments and aim to address them now.
> ___
> # Justification of our DR estimator in the context of other estimators
> We thank the reviewer for pointing us towards these works and acknowledge that they do provide robust methods for estimating the CQTE. We will make sure to include these works in our paper and discuss the tradeoffs between them.
>
> We see the main strength of our work being the fact that the CQC is a more desirable object to target than the CQTE. This is because the smoothness of the CQC appears to be a more intuitive and naturally occurring phenomenon than the smoothness of the CQTE. Example 1 in our paper illustrates this: the CQTE still contains high-frequency oscillation terms, while the CQC does not. We have attached Figure 2, showing the "non-smoothness" of the CQTE, to the PDF and will include it in the final paper.
>
> In general, there seems to be limited intuition about what smoothness of the CQTE represents (for example, there is not much discussion of this in Kallus et al. (2023) paper). This contrasts with the CQC, where we believe there is a more intuitive explanation. Specifically, if we are in the potential outcome framework and $Y_1=\phi(Y_0,X)$ with $\phi$ being some increasing function in $Y_0$, then $\phi$ is the CQC. Thus, smoothness in this $\phi$ represents smoothness in the CQC. This provides a generalisation to the CATE case, where we observe smoothness of the CATE when $Y_1=Y_0+\phi(X)$ with $\phi(X)$ being smooth. We naturally therefore see the CQC as the more general extension of the CATE when comparing two distributions (note: here the CATE transformation is also monotonic in $Y_0$.)
> A simple example of this would be a treatment that halves the response for an individual (e.g. a blood pressure medication which halves an individual's blood pressure.) In this case the CQC would be $g^*(y,x)=0.5y$ whereas the CQTE would depend upon the distribution of $Y|X,0$ making it non-smooth if this marginal distribution is non-smooth.
>
>
> Outside this potential outcome framework we can also think of smoothness in the CQC as representing a smooth change in the conditional QQ-plot between $Y|X=x, A=0$ and $Y|X=x,A=0$ as you vary the covariates ($x$).
> Overall, we feel the CQC is a natural way to characterise the relationship between two distributions, having many useful alternative definitions. For example, the CQC is the unique monotonic function that maps one distribution to the other. This monotonicity seems an intuitive restriction, as it ensures that healthier patients without treatment are mapped to healthier patients under treatment.
> ___
> # Comparing estimation procedure tradeoffs
> We appreciate your comment about discussing the trade-offs between our approach and that of Kallus et al. (2023) and will make sure to do so. We agree that our approach is more statistically intensive due to the need to invert our CCDF contrasting function. In terms of complexity, while our procedure does have more steps, it has the benefit of estimating fewer nuisance parameters. Specifically, we do not need to estimate the reciprocal conditional PDFs, which can be difficult to estimate in low-density regions where errors can blow up. Even with the desirable second-order error terms that this estimation incurs, it could still have a strongly negative impact on the estimation accuracy.
> ___
> # Generalising beyond Linear Smoothers
> We appreciate the reviewers' suggestion to extend our results beyond linear smoothers. In the context of Kallus et al. (2023), our result represents a different assumption about the class of regression functions, rather than a more restrictive one.
>
> Linear smoothers encompass a broad range of methods, including kernel ridge regression, k-NN regression, Mondrian forests, and generalised random forests (refer to the global response for details and references). Additionally, Theorem 3 of Kallus et al. (2023) incorporates Kennedy's (2023) stability condition (acknowledged in their proof in Appendix A4). Since stability has primarily been demonstrated for linear smoothers (in Kennedy (2023)), this result is comparable to ours.
>
> Our requirement for CCDF estimators to be linear smoothers in Section 3.4 is solely to control the maximum step size. We are open to replacing this with a more general condition on the step size, which would mean only requiring the outer regression to be a linear smoother.
>
> While Kallus et al. (2023) provide Theorem 2 with desirable convergence rates beyond linear smoothers, their assumptions on bracketing entropy restrict function classes one can use significantly. Therefore, we view this as an alternative result rather than a more general one. Since submitting the paper, we have obtained preliminary results in this framework and are prepared to include them in the supplementary material.
>
> Finally, a key advantage of our result is the provision of pointwise accuracy bounds, as opposed to bounds in expectation over responses and covariates. This allows our accuracy results to adapt to local smoothness and avoids penalties for the worst points.
> ___
> # Comparing estimation results
> We have taken your suggestion and compared our method against theirs in our empirical results. This can be seen in Figure 1 of the PDF document.
>
> The results show that our approach (green/purple) clearly outperforms the CQTE estimation approach (brown) as the frequency term ($\gamma$) increases in the left plot. This is because the CQTE has a complexity that depends on the frequency term, while the CQC does not. We also observe much better performance as the sample size increases. We hypothesise that the plateau in the CQTE approach is due to the difficulty of estimating the reciprocal of the PDF, which causes the estimation to be unstable regardless of sample size.
> ___
> # References
> Kennedy, E. H. (2023). Towards optimal doubly robust estimation of heterogeneous causal effects. Electronic Journal of Statistics

---

> > ### Comment · Reviewer_nAy3 · 2024-08-12
> >
> > Thanks for your response. My concernes have been addressed. I have raised my score with the expectation that the authors will include this discussion (especially the comparisons to Kallus et al. (2023)) in their final version.

---

> > > ### Author Response · Authors · 2024-08-13
> > >
> > > Thank you for the reply. We will make sure to include the comparisons with Kallus et al (empirically, methodologically, and theoretically) in the final paper.

---

### Official Review · Reviewer_bJi9 · 2024-07-17

**Soundness:** 3
**Presentation:** 3
**Contribution:** 2
**Rating:** 6
**Confidence:** 2

**Summary:**

The authors propose a new estimand called the Conditional Quantile Comparator (CQC), which computes the quantiles of the outcome distributions of the treatment effect, hence improving on the (usual) conditional average treatment effect. The CQC is defined as a measurable function that maps an untreated outcome to the equivalent treated outcome in the same quantile, conditional on covariates. The authors introduce a doubly robust estimation procedure for CQC, where the estimation is re-framed as a CATE problem using the pseudo-outcome framework. The authors present finite sample bounds on the estimator's error, and provide numerical simulations and result on a real employment dataset.

**Strengths:**

- The paper is well written and easy to follow
- The paper provides a thorough theoretical analysis for the finite sample bounds and proofs of convergence rates for CQC and proposed estimation algorithms
- The simulations and real use-case considered provide a good intuition to the reader on the benefits of the proposed approach

**Weaknesses:**

First of all, I'd like to say I am not very familiar with the causal inference literature, so some of my comments might not be as relevant.

- Complexity of using quantiles: can the authors comment in terms of error bounds and convergence rates what is the price to pay to estimate quantiles as opposed to CATE? Some of these results, including the use of pseudo-outcomes and isotonic projections, appear a little bit unusual, as usually quantiles are estimated through the pinball loss. As usual what is important is assessing the left tail or right tail (i.e., whether the treatment effect crosses the zero or not), I would expect the right/left tail quantiles to be the most important ones -- is the error in estimating all the quantiles the same across the entire distribution?

- Dimensionality of the input $\textbf{x}$: both the simulation and the real dataset use a very low-dimensional $\textbf{x}$, and the comments on using the Nadaraya-Watson estimator and kernel smoothing are really only applicable when X is generally low-dimensional. How does in practice your method behave as function of the input dimension $\textbf{x}$, and would your comments still be applicable even in higher-dimensional settings?

- Estimating quantiles vs entire distribution: how does this approach compare to other approaches that estimate the entire conditional density, such as [1]? Is the use of a doubly robust framework guarantee the optimal error rates?


[1] Zhou, T., Carson IV, W. E., & Carlson, D. (2022). Estimating potential outcome distributions with collaborating causal networks. Transactions on machine learning research, 2022.

**Questions:**

See the weaknesses section above.

**Limitations:**

The authors do address some limitations and implications of their approach.

---

> ### Author Rebuttal · Authors · 2024-08-07
>
> We thank the reviewer for their detailed and salient comments and aim to address them now.
> ___
> # Complexity of using quantiles
> Regarding the indirect estimation of the quantiles: due to the monotonicity of the CCDFs and $h^*$, we incur only a trivial increase in our error bounds for estimating the CCDF and then inverting when compared to directly estimating the quantiles.
> Specifically, our rate only gains a $\log(n)$ factor. This trivial cost is greatly outweighed by the benefit of being able to properly exploit the smoothness of our estimand.
>
> The isotonic projection step itself is a fairly minor technical detail to ensure our desirable theoretical results. Empirically, we find the function to be close to monotonic before the projection. In Figure 1, we plot our approach with and without isotonic projection (taking the generalized inverse in both cases) and see no meaningful difference in the outcome (the green and purple lines).
>
> When it comes to estimating specific tails of the distribution, our method allows us to specify an input $y_0$ for our CQC, enabling us to focus on the tails if desired. We do have to estimate across all $y_1$ values, however as mentioned, this does not pose an issue regarding our estimation accuracy.
> ___
> # Dimensionality of the Input
> We appreciate that NW estimation may not generalise to high dimensions quite as well, however, our theory is for the more general class of linear smoothers. These encompass many methods (k-NN regression, Mondrian forests, Generalised forests), including methods associated with high-dimensional data such as kernel ridge regression (see global response for references and more detail). We use NW estimation as a simple illustrative example as we focus more on the general idea of the pseudo-outcome regression rather than attempting to optimise the regression technique itself. Furthermore, as we are assuming simplicity of the $h^*$, we can exploit the fact that some linear smoothing approaches can adapt to simple/low-dimensional manifolds in our covariates even when the data itself is high-dimensional.
>
> Finally, while our theory is for the case of linear smoothers, in practice, it can extend beyond this, and other high-dimensional regression techniques can be used.
> ___
> # Estimating quantiles vs Entire Distribution
> Thank you for bringing this work to our attention, we will make sure to mention it in our paper.
> In Zhou et al., the output is separate estimates of the two conditional distributions ($Y|X,A=0$ and $Y|X,A=1$).  As such, the optimal rates for estimating these distributions will depend on the smoothness of the marginal quantiles. In our approach, we show that we are able to obtain rates that are better than the estimation rates for the marginal quantiles, and thus necessarily improve upon the estimation rates obtained in their work.
>
> We have also added a comparison to another work on robust CQTE estimation (Kallus et al., 2023). This can be seen in Figure 1, where we achieve better performance (green/purple line) compared to the robust CQTE method (brown line). This improvement is due to the smoothness present in the CQC, which is not present in the CQTE or the marginal distributions.
> ___
> # References
> Kallus, N. and Oprescu, M. (2023). Robust and agnostic learning of conditional distributional treatment effects. In Ruiz, F., Dy, J., and van de Meent, J.-W., editors, Proceedings of the 26th international conference on artificial intelligence and statistics, volume 206 of Proceedings of machine learning research, 6037–6060. PMLR.
>
> Zhou, T., Carson, W. E. t., and Carlson, D. (2022). Estimating Potential Outcome Distributions with
> Collaborating Causal Networks. Transactions on machine learning research, 2022. Place: United States.

---

> > ### Comment · Reviewer_bJi9 · 2024-08-11
> >
> > I thank the authors for their comments. In particular, please include the discussion about the complexities of using quantiles in the paper, as it was not clear to me that inverting the CCDF is actually not that bad in terms of an idea in practice and in terms of error rate. Due to the author points in this response and in the global rebuttal, I increase my score within the limits of my understanding and experience with the subject.

---

> > > ### Author Response · Authors · 2024-08-12
> > >
> > > Thank you for the reply. We will make sure to include a further discussion on the overall estimation difficulty of the CCDF when compared to quantile estimation in the final paper.

---

### Author Rebuttal · Authors · 2024-08-07

We would like to thank all of the reviewers for their thoughtful questions and valuable feedback. We address most points in the reviewer-specific responses but will highlight a few key points here as well.
___

# Comparisons to other approaches
In line with feedback from multiple reviewers, we have included a comparison to a CQTE estimation approach in our empirical results. This new comparison can be seen in Figure 1 of the attached PDF. The approach from Kallus et al. (2023) is represented in brown in the figure. As shown, it performs significantly worse than our approach, which we believe is due to the increased complexity of the CQTE in this example compared to the CQC. To compare this estimator with CQC estimators, we used the transformation highlighted in Section 2.2 of our paper to produce a CQC estimator. This transformation was applied using the exact CDFs to ensure that it was not the transformation affecting performance.
___

# Linear Smoothers and NW estimation
Multiple reviewers mentioned the potentially poor performance of NW estimation in higher dimensions. While this is absolutely true, we are using NW estimation as a simple example of a linear smoother for illustrative purposes in our experiments. Our theory itself applies to all linear smoothers. Despite their simplistic definition, linear smoothers encompass a deceptively broad class of estimation techniques used in both low and high-dimensional settings. Examples include k-NN regression (Chen, 2019; Chen et al., 2019), kernel ridge regression (Singh et al., 2023), generalised forests (Athey et al., 2019), and Mondrian forests (Lakshminarayanan et al., 2014). Additionally, even in high-dimensional cases, linear smoothers have been shown to adapt to intrinsic low dimensionality, which relates to our notion of the CQC being simpler or smoother than the marginal quantities (Kpotufe, 2011).
___

# References
Athey, S., Tibshirani, J., and Wager, S. (2019). Generalized random forests. The Annals of Statistics,
47(2):1148 – 1178. Publisher: Institute of Mathematical Statistics.

Chen, G. (2019). Nearest neighbor and kernel survival analysis: Nonasymptotic error bounds and strong
consistency rates. In Chaudhuri, K. and Salakhutdinov, R., editors, Proceedings of the 36th international
conference on machine learning, volume 97 of Proceedings of machine learning research, pages 1001–1010. PMLR.

Chen, P., Dong, W., Lu, X., Kaymak, U., He, K., and Huang, Z. (2019). Deep representation learning for in-
dividualized treatment effect estimation using electronic health records. Journal of Biomedical Informatics,
100:103303.

Kallus, N. and Oprescu, M. (2023). Robust and agnostic learning of conditional distributional treatment
effects. In Ruiz, F., Dy, J., and van de Meent, J.-W., editors, Proceedings of the 26th international
conference on artificial intelligence and statistics, volume 206 of Proceedings of machine learning research,
pages 6037–6060. PMLR.

Kpotufe, S. (2011). k-NN regression adapts to local intrinsic dimension. In Shawe-Taylor, J., Zemel, R.,
Bartlett, P., Pereira, F., and Weinberger, K., editors, Advances in neural information processing systems,
volume 24. Curran Associates, Inc.

Leqi, L. and Kennedy, E. H. (2022). Median optimal treatment regimes. arXiv: 2103.01802 .

Singh, R., Xu, L., and Gretton, A. (2023). Kernel methods for causal functions: dose,
heterogeneous and incremental response curves. Biometrika, 111(2):497–516.

---

### Decision · Program_Chairs · 2024-09-25

**Decision:**

Accept (poster)

**Comment:**

This work proposes a new estimand for comparing the quantiles between control and treatment groups. The Conditional Quantile Comparator (CQC) maps an outcome from the control group to an outcome in the treatment group, such that they represent the same conditional quantiles in their respective distributions. The paper is motivated by the shortcomings of the Conditional Quantile Treatment Effect (CQTE), which is not robust to errors in the quantiles.

The authors propose an doubly-robust estimation procedure which estimates a conditional contrasting function that examines the difference between conditional CDFs, and then performs isotonic regression. The contrasting function is estimated using pseudo-outcome regression, which provides double robustness.

Several experts reviewed the paper and raised concerns on interpretability and lack of appropriate comparisons to prior work. The authors have addressed these issues during the discussion period, which should be incorporated in the camera-ready version of the paper.